# Endosomal TLR3 co-receptor CLEC18A enhances host immune response to viral infection

Ya-Lang Huang[1], Ming-Ting Huang[1], Pei-Shan Sung [1], Teh-Ying Chou[2,3], Ruey-Bing Yang [4], An-Suei Yang[1], Chung-Ming Yu[1], Yu-Wen Hsu[5], Wei-Chiao Chang[5] & Shie-Liang Hsieh [1,3,6,7✉]

Human C-type lectin member 18A (CLEC18A) is ubiquitously expressed in human, and highest expression levels are found in human myeloid cells and liver. In contrast, mouse CLEC18A (mCLEC18A) is only expressed in brain, kidney and heart. However, the biological functions of CLEC18A are still unclear. We have shown that a single amino acid change ($S_{339}$ →$R_{339}$) in CTLD domain has profound effect in their binding to polysaccharides and house dust mite allergens. In this study, we further demonstrate that CLEC18A and its mutant CLEC18A(S339R) associate with TLR3 in endosome and bind poly (I:C) specifically. Compared to TLR3 alone, binding affinity to poly (I:C) is further increased in TLR3-CLEC18A and TLR3-CLEC18A(S339R) complexes. Moreover, CLEC18A and CLEC18A(S339R) enhance the production of type I and type III interferons (IFNs), but not proinflammatory cytokines, in response to poly (I:C) or H5N1 influenza A virus (IAV) infection. Compared to wild type (WT) mice, ROSA-CLEC18A and ROSA-CLEC18A(S339R) mice generate higher amounts of interferons and are more resistant to H5N1 IAV infection. Thus, CLEC18A is a TLR3 co-receptor, and may contribute to the differential immune responses to poly (I:C) and IAV infection between human and mouse.

[1] Genomics Research Center, Academia Sinica, Taipei, Taiwan. [2] Department of Pathology, Taipei Veterans General Hospital, Taipei, Taiwan. [3] Institute of Clinical Medicine, National Yang-Ming University School of Medicine, Taipei, Taiwan. [4] Institute of Biomedical Sciences, Academia Sinica, Taipei, Taiwan. [5] Department of Clinical Pharmacy, School of Pharmacy, Taipei Medical University, Taipei, Taiwan. [6] Department of Medical Research and Education, Taipei Veterans General Hospital, Taipei, Taiwan. [7] Institute for Cancer Biology and Drug Discovery, Taipei Medical University, Taipei, Taiwan. ✉email: slhsieh@gate.sinica.edu.tw

Human C-type lectin 18 (clec18) gene cluster contains three loci clec18a, clec18b, and clec18c, and each clec18 family member encodes a polypeptide of 446 amino acids comprising a C-type lectin-like domain (CTLD) in the C terminus and a sperm-coating protein (SCP) domain (also known as CAP or TAPS domain) in the N terminus[1]. The three human C-type lectin member 18A/B/C (CLEC18A/B/C) polypeptides are almost identical except polymorphic amino acid residues located in CTLD and SCP/CAPT/APS domains[1]. Human CLEC18A is ubiquitously expressed in human tissues with higher expression levels in myeloid cells and liver. However, the biological function of CLEC18A is still unknown, nor do we understand the impact of polymorphic amino acid residues in CTLD.

It has been shown that endosomal Toll-like receptors (TLRs) recognize diverse danger signals and nucleic acid components released from uncoated virions, and activation of endosomal TLRs induces the production interferons (IFNs) and proinflammatory cytokines to initiate inflammatory responses[2]. Endosomal TLR3 is regarded as a sentinel to RNA viruses, and activation of TLR3 leads to TRIF (TIR domain-containing adaptor-inducing IFN-β)-dependent IFN production and nuclear factor-κB (NF-κB)-dependent proinflammatory cytokine production[3]. Interestingly, the differential outcome of TLR3-mediated signaling between human and mouse have been noted[4,5]. Compared to wild-type (WT) mice, TLR3$^{-/-}$ mice produced lower levels of inflammatory mediators and displayed less cell infiltration after H3N2 influenza A virus (IAV) infection, despite higher virus titer being noted in the lung[6]. This observation suggests that TLR3-mediated signaling prefers to induce proinflammatory cytokine release and has a detrimental effect in IAV infection in mice. In contrast, human TLR3 is as important as RIG-I in the production of type I and type III IFNs after IAV infection[7], and intranasal administration of TLR3 agonist reduced lung viral titers and mortality in mice[8]. These observations suggest that TLR3 activation is able to produce IFNs to protect the host from IAV infection. However, the molecular mechanism responsible for the differential TLR3-mediated signaling between human and mouse is still unknown.

Since polymorphic amino acid residues were found in the CTLD domain and SCP domain among members of CLEC18 family, and distinct glycan-binding affinity in vitro was found between the CTLD domain of CLEC18A(S339) and CLEC18A-1(R339)[1], we are interested to understand the differential functions of S339 versus R339 in host immunity in vivo. To address this question, we generate CLEC18A(S339R) mutant, which is identical to CLEC18A except amino acid 339 (S→R). Here we reported that CLEC18A and CLEC18A(S339R) associate with TLR3 and bind poly (I:C) directly. To understand their roles in TLR3-mediated signaling, we generated ROSA-CLEC18A and ROSA-CLEC18A(S339R) mice to investigate their functions in poly (I:C) stimulation and IAV infection. We found that macrophages and alveolar epithelial cells (AECs) from ROSA-CLEC18A and ROSA-CLEC18A(S339R) mice produce higher amounts of IFNs (type I and type III) and IFN-stimulated genes (ISGs) after incubation with poly (I:C) or H5N1 IAV. Furthermore, ROSA-CLEC18A and ROSA-CLEC18A(S339R) mice are more resistant to H5N1 infection with higher survival rates than WT littermates (CLEC18A(S339R) > CLEC18A > WT). These observations suggest that CLEC18A and CLEC18A(S339R) act as TLR3 co-receptors in the endosome, and may contribute to the differential immune responses to TLR3-mediated signaling between human and mouse.

## Results

### Expression of mouse endogenous CLEC18A and generation of ROSA-CLEC18A/ CLEC18A(S339R) knock-in mice. While

human CLEC18A is ubiquitously expressed and highest expression levels were noted in immune cells and liver[1], mouse CLEC18A (mCLEC18A) messenger RNA (mRNA) is only expressed in brain, kidney, and heart (Fig. 1a). To confirm the expression of mCLEC18A, we generated human anti-mCLEC18A monoclonal antibody (mAb) to detect mCLEC18A in mouse tissues (lanes 1–8, Fig. 1b). We harvested cell lysates from 293T cells transfected with vector alone (lane 9, Fig. 1b) or Flag-tagged full-length mCLEC18A (lane 10, Fig. 1b), followed by fractionation on sodium dodecyl sulfate-polyacrylamide gel electrophoresis (SDS-PAGE) before blotting to polyvinylidene difluoride (PVDF) paper. The blots were incubated with anti-mCLEC18A mAb (lanes 9 and 10, Fig. 1b) and anti-Flag mAb (lanes 9 and 10, Fig. 1c), respectively, to determine the expression of mCLEC18A. We found that anti-mCLEC18A mAb is able to detect endogenous mCLEC18 in the brain (lane 1, Fig. 1b) and 293T cells transfected with full-length mCLEC18A (lane 10, Fig. 1b), while the expression levels of mCLEC18A in heart and kidney (lanes 3 and 7, Fig. 1b) were too low to be detected under the same condition. We further examined the expression of endogenous mCLE18A by flow cytometry (Supplementary Fig. 1). Even we can detect 293T cells transfected with full-length mCLEC18A complementary DNA (cDNA) (Supplementary Fig. 1d), we cannot detect mCLEC18A in mouse peripheral blood, bone marrow, and splenocytes. These observations further confirmed that the tissue distribution of mCLEC18A is distinct from the human counterpart.

To understand the functions of CLEC18A-mediated immune responses in vivo, we generated ROSA-CLEC18A and ROSA-CLEC18A(S339R) mice to express human CLEC18A and CLEC18A(S339R), respectively, to address this question. We found that CLEC18A and CLEC18A(S339R) were expressed ubiquitously, and the highest expression level was found in the brain, bone marrow, and splenocytes in ROSA-CLEC18A and ROSA-CLEC18A(S339R) mice (Fig. 1d). The expression of human CLEC18A and CLEC18A(S339R) in ROSA-CLEC18A and ROSA-CLEC18A(S339R) mice tissues is similar to what we observed in human tissues[1], although the expression levels are lower than that in human liver and kidney. Western blot analysis further confirmed the expression of CLEC18A and CLEC18A (S339R) in mouse bone marrow-derived macrophages (BMDMs) and dendritic cells (BMDCs) (Fig. 1e).

### CLEC18A enhances IFN production and ISG expression after influenza virus infection. As mouse BMDMs do not express endogenous CLEC18A, we compared the responses of BMDMs isolated from WT, ROSA-CLEC18A, and ROSA-CLEC18A (S339R) mice to various TLR ligands, respectively, followed by examining the expression of type I IFNs (IFN-α and IFN-β) and proinflammatory cytokines. Among the TLR ligands tested, high molecular weight (HMW) poly (I:C) enhanced the expression of IFN-α, IFN-β, IFN-γ-inducible protein 10 (IP-10), interleukin-6 (IL-6), and tumor necrosis factor-α (TNF-α). Interestingly, the expression levels of IFN-α, IFN-β, and IP-10 were further upregulated in ROSA-CLEC18A and ROSA-CLEC18A(S339R) BMDMs (Fig. 2a). Similar to poly (I:C) stimulation, higher expression levels of IFN-α and IFN-β were found in ROSA-CLEC18A and ROSA-CLEC18A(S339R) BMDMs stimulated with RNA viruses, including H5N1 IAV and dengue virus (DV) (Fig. 2b). In contrast, other TLR ligand has no obvious effect to upregulate cytokine production (Supplementary Fig. 2). We further knocked down the endogenous CLEC18 in human macrophage to confirm its role in TLR3-mediated signaling. We found that knockdown of CLEC18 significantly reduced poly (I:C) and virus-, but not LPS-, induced IFN production (Supplementary

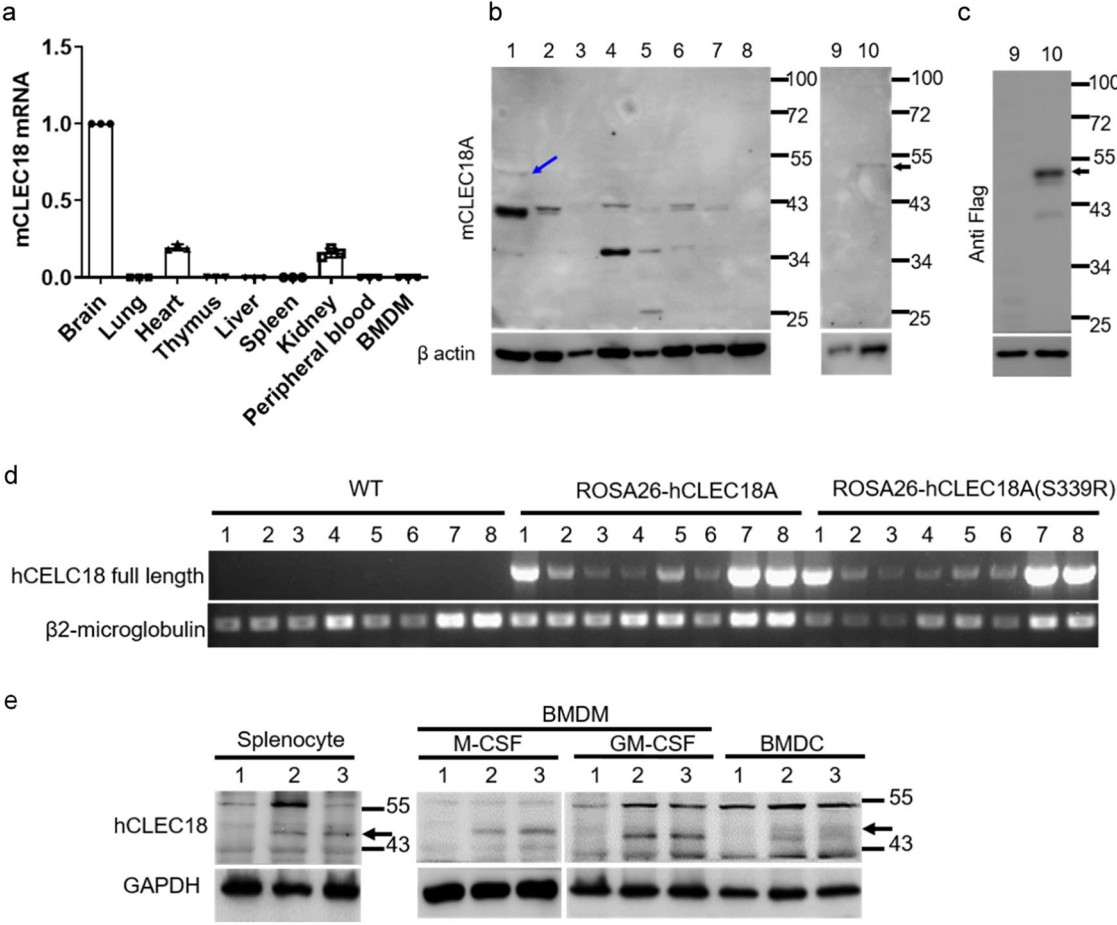

**Fig. 1 Expression mouse endogenous CLEC18A and generation of ROSA- CLEC18A/CLEC18A(S339R) knock-in mice. a–c** Expression of mouse endogenous CLEC18A (mCLEC18) mRNA and protein in B6 mice was determined by RT-PCR (**a**) and WB (**b, c**). 1: Brain, 2: Lung, 3: Heart, 4: Thymus; 5: Liver, 6: Spleen, 7: Kidney, 8: Bone marrow; 9: 293T cells transfected with pCMV-Tag4A (vector control); 10: 293T cells transfected with pCMV-Tag4A-mCLEC18 (50 kDa). Arrow indicates the position of mouse CLEC18 ($n = 3$). **d** Expression of human CLEC18A and CLEC18A (S339R) in ROSA-CLEC18A and CLEC18A(S339R) mice was determined by PCR. 1: Brain; 2: Lung; 3: Heart; 4: Liver; 5: Spleen; 6: Kidney; 7: Bone marrow; 8: Splenocyte ($n = 3$). **e** Detection of human CLEC18 protein (50 kDa) in mouse bone marrow-derived dendritic cells (BMDCs), as well as macrophages (BMDMs) cultured under M-CSF and GM-CSF, respectively. 1: Wild type mice (WT); 2: ROSA26-CLEC18A mice; 3. ROSA-CLEC18A(S339R) mice. Arrow indicates the molecular weight (50 kDa) of human CLEC18A/CLEC18A(S339R) (hCLEC18) ($n = 3$).

Fig. 3). To further examine the role of CLEC18A and CLEC18A (S339R) in TLR3-mediated signaling, human HT1080 cells were transfected by small interfering RNAs (siRNAs) to knock down endogenous CLEC18A. After recovery overnight, the transfected cells were transfected with pMACS.Kk-HA(C)-hCLEC18A and pMACS.Kk-HA(C)-hCLEC18A(S339R) to express CLEC18A and CLEC18A(S339R), respectively. We found that overexpression of CLEC18A and CLEC18A(S339R) enhanced the expression of IFN-α and IP-10 after poly (I:C) stimulation (Supplementary Fig. 4). These observations further confirm that CLEC18A is able to enhance TLR3-mediated signaling. Moreover, we examined the expression of ISGs critical in viral infection[9,10]. While Ch25h and Mx1 were upregulated, the expression of ISG15 was similar in both ROSA-CLEC18A and ROSA-CLEC18A(S339R) BMDMs post H5N1 IAV and DV infections (Fig. 2c). In addition, virus titer and the expression of matrix protein (M) and nucleoprotein (NP) were downregulated in ROSA-CLEC18A and CLEC18A (S339R) BMDMs (Fig. 2d). It is interesting to note that CLEC18A (S339R) is more potent than CLEC18A to enhance IFN production and suppress virus replication in all the assays.

In addition to type I IFNs, we further investigated the expression of type III IFN, which plays a critical role in anti-influenza virus infection[11], in AECs after H5N1 IAV infection.

Compared to WT mice, both type I IFNs (IFNα4 and IFNβ) and type III IFN (IFNλ2/3) were upregulated, while the expression of TNF-α and IL-6 was similar in AECs isolated from ROSA-CLEC18A and ROSA-CLEC18A(S339R) mice (Fig. 3a). Moreover, the expression of M and NP genes was suppressed in AECs expressing human CLEC18A and CLEC18A(S339R) (Fig. 3b). Similar to what we observed in macrophages, CLEC18A(S339R) is more potent than CLEC18A to induce IFN expression and suppress virus replication in virus-infected AECs, suggesting that the polymorphic amino acid residue [$S_{339} \rightarrow R_{339}$] in CTLD domain has a profound effect to upregulate IFN production and suppress H5N1 IAV replication.

**CLEC18A and CLEC18A(S339R) associate with TLR3 and bind poly (I:C) directly.** Because CLEC18A is located in the endosome, we asked whether CLEC18A associated with TLR3 in the presence of poly (I:C) or H5N1 IAV virus, respectively. We found that HMW poly (I:C) and H5N1 IAV increased the efficiency of fluorescence resonance energy transfer (FRET) between TLR3-CLEC18A, respectively (Fig. 4a, b). Because TLR3 is expressed in early endosome (pH 6.3)[12,13], we further asked whether CLEC18A.Fc and CLEC18A(S339R).Fc bind poly (I:C) directly by

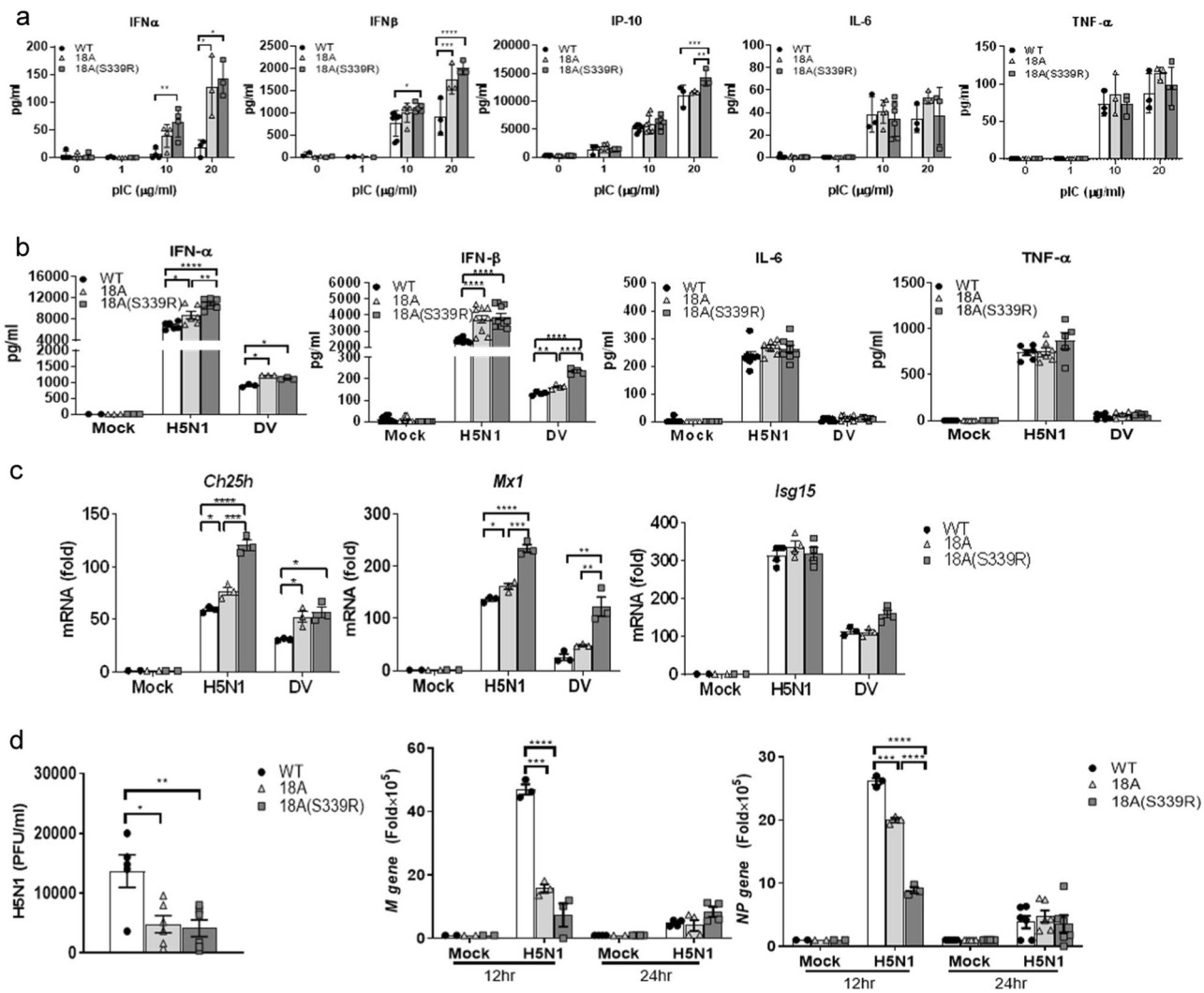

**Fig. 2 CLEC18 enhances IFN production and ISG expression to suppress virus replication. a** BMDMs (1 × 10⁶/well) from WT, ROSA-CLEC18A, and ROSA-CLEC18A(S339R) mice were incubated with HMW poly (I:C) for 24 h, followed by harvesting cells to determine the expression of cytokines by ELISA (*n* = 3–6). **b, c** BMDMs (1 × 10⁶/well) from WT, ROSA-CLEC18A, and ROSA-CLEC18A(S339R) mice were incubated with RNA viruses (H5N1 MOI = 2; DV MOI = 30) for 24 h, followed by harvesting cells to determine cytokines expression by ELISA (*n* = 3–10) (**b**) and IFN-stimulating genes (ISG) expression by quantitative RT-PCR (*n* = 3) (**c**). **d** Virus replication was determined by plaque assay (left), and quantitative RT-PCR to determine the expression of virus structure genes (M) and nucleoprotein (NP) (right) (*n* = 3–10). One-way ANOVA was performed. *P < 0.05, **P < 0.01, ***P < 0.001 and ****P < 0.0001, for WT group versus group of CLEC18A or CLEC18A(S339R). H5N1 type A influenza virus, DV dengue virus.

bio-layer interferometry at pH 6.3. We found that CLEC18A (S339R).Fc bound HMW poly (I:C) ($K_D$ = 8.17E − 09) with similar affinity as TLR3.Fc ($K_D$ = 5.59E − 09), whereas the binding affinity of CLEC18A.Fc to poly (I:C) ($K_D$ = 2.10E − 08) is slightly lower than CLEC18A(S339R).Fc under the same condition (Supplementary Table II). Interestingly, co-immobilization of TLR3.Fc with CLEC18A.Fc or CLEC18A(S339R).Fc, respectively, further increases binding affinity to HMW poly (I:C). The affinity between poly (I:C) and TLR3.Fc/ CLEC18A(S339R) ($K_D$ ≤ 1.0E − 12) is much higher than TLR3.Fc alone ($K_D$ = 5.59E − 09), while TLR3.Fc/CLEC18A ($K_D$ = 4.46E − 10) is slightly higher than TLR3.Fc alone. Co-localization of human CLEC18A with human and mouse TLR3 in early endosome was also observed in human macrophage (Fig. 4c) and mouse BMDMs from ROSA-CLEC18A/ CLEC18A(S339R) mice (Fig. 4d). To further confirm the association between CLEC18 and TLR3 contributes to enhanced IFN production, we added the TLR3/dsRNA (double-stranded RNA) complex inhibitor, which inhibited IFN production via competing HMW poly (I:C) binding

to TLR3, to disrupt the TLR3/CLEC18/dsRNA complex. We found that IFN production was reduced by the TLR3/dsRNA complex inhibitor in a dose-dependent manner (from 30 to 100 μM) in WT BMDM. In contrast, IFN production and the efficiency of FRET between TLR3-CLEC18A/ CLEC18A(S339R) was not reduced in lower doses (10 and 30 μM) of TLR3/dsRNA complex inhibitor until it reaches 100 μM in BMDMs isolated from ROSA-CLEC18A and ROSA-CLEC18A(S339R) mice (Fig. 5a, b). These observations suggest that the binding affinity between HMW poly (I:C) and TLR3-CLEC18A/CLEC18A (S339R) complexes were increased, thereby requiring a higher concentration of inhibitor to suppress HMW poly (I:C)-induced cytokine production. Furthermore, enhanced phosphorylation of TBK1 and IRF3 was observed in BMDMs isolated from CLEC18A and CLEC18A(S339R) mice (Fig. 5c). Thus, we conclude that CLEC18A and CLEC18A(S339R) act as TLR3 co-receptor in the endosome, thereby enhancing TLR3-mediated signaling and IFN production after incubation with HMW poly (I:C).

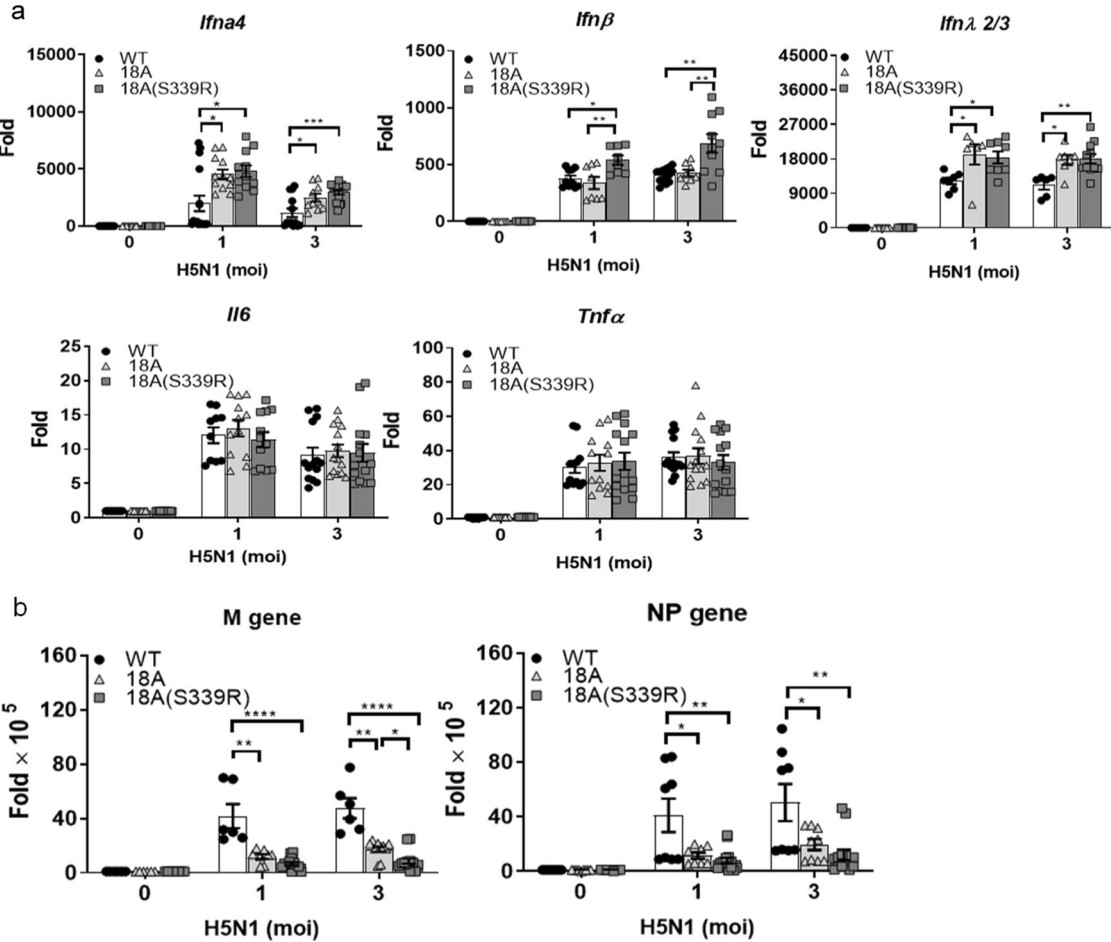

**Fig. 3 CLEC18A and CLEC18A(S339R) modulate type I/III IFN expression and virus replication in lung alveolar epithelial cells.** Pulmonary alveolar epithelial cells ($2 \times 10^5$/well) from WT, ROSA-CLEC18A, and ROSA-CLEC18A(S339R) mice were incubated with H5N1 virus (MOI = 2) for 12 h, followed by harvesting cells to determine cytokine expression (**a**) and virus replication (**b**) by quantitative RT-PCR ($n = 7$–12). One-way ANOVA was performed. *$P < 0.05$, **$P < 0.01$, ***$P < 0.001$ and ****$P < 0.0001$, for WT group versus group of CLEC18A or CLEC18A(S339R).

**Mapping of TLR3 and CLEC18A/ CLEC18A(S339R) interaction domain.** We further identified the interaction domains between TLR3 and CLEC18A/CLEC18A(S339R) by immunoprecipitation using TLR3 deletion mutants (Fig. 6a). We found that CLEC18A and CLEC18A(S339R) associated with TLR3 deletion mutant without Toll/IL-1 receptor homology domain (TLR3△TIR), but they were unable to associate with TLR3 deletion mutant without leucine-rich repeat (LRR) domain (TLR3△LRR) (Fig. 6a–c). It has been reported that poly (I:C) interacts with TLR3 via C-terminal LRR20 and N-terminal-binding site LRR1[14,15]. Therefore, we generated TLR3△LRR$_{1-3}$ (TLR3 without 1–3 LRR repeats) and TLR3△LRR$_{19-22}$ (TLR3 without 19–22 LRR repeats) to map the minimal region of TLR3 responsible for interacting with CLEC18A and CLEC18A(S339R). As shown in Fig. 6d–g, TLR3 mutants (TLR3△LRR$_{1-3}$ and TLR3△LRR$_{19-22}$) still interacted with CLEC18A and CLEC18A(S339R), respectively. This observation suggests that TLR3 interacted with CLEC18A and CLEC18A(S339R) via LRR$_{4-18}$, which is not overlapping with poly (I:C) interacting LRR1 and LRR20 of TLR3.

We further generated CLEC18A deletion mutants to identify minimal domain responsible for interaction with TLR3. We found that all the CLEC18 deletion mutants interact TLR3 efficiently except CLEC18 Δ28–292 a.a. (Fig. 6h–k). Therefore, we conclude that CTLD domain is dispensable for CLEC18A–TLR3 interaction, and TLR3 interacts with all the rest parts of CLEC18A, including CAP-, EGF- and EGK-like domains.

**CLEC18 enhances poly (I:C)-induced inflammatory reaction in vivo.** We further asked whether CLEC18A and CLEC18A (S339R) enhanced HMW poly (I:C)-induced inflammatory reaction in vivo. To address this question, mice were inoculated with HMW poly (I:C) via intranasal route, and then were sacrificed at days 1 and 4 post inoculation to examine lung inflammation by histochemical staining. Compared to WT littermates, severe cell infiltration was noted in ROSA-CLEC18A and ROSA-CLEC18A(S339R) mice (CLEC18A(S339R) > CLEC18A > WT) at days 1 and 4 post inoculation (Fig. 7a). We found that production of type I IFNn (IFN-α, IFN-β), type III IFN (IFNλ2/3), and IP-10 was upregulated in ROSA-CLEC18A(S339R) and ROSA-CLEC18A mice (Fig. 7b). This observation further supports the notion that CLEC18A(S339R) is more potent than CLEC18A to enhance poly (I:C)-induced inflammation.

**CLEC18A and CLEC18A(S339R) protect host from H5N1 IAV-induced inflammation and lethality.** We further examined the effect of CLEC18A and CLEC18A(S339R) in host defense against H5N1 IAV infection. To address this question, mice were inoculated with H5N1 IAV (1500 PFU/mouse) via intranasal route to observe body weight change and survival rate for 21 days. We found that CLEC18A(S339R) is more potent than CLEC18A to protect mice from H5N1 IAV-induced body weight loss (Fig. 8a) and lethality (Fig. 8b). Compared to WT mice, less cell infiltration and pulmonary inflammation were noted in ROSA-

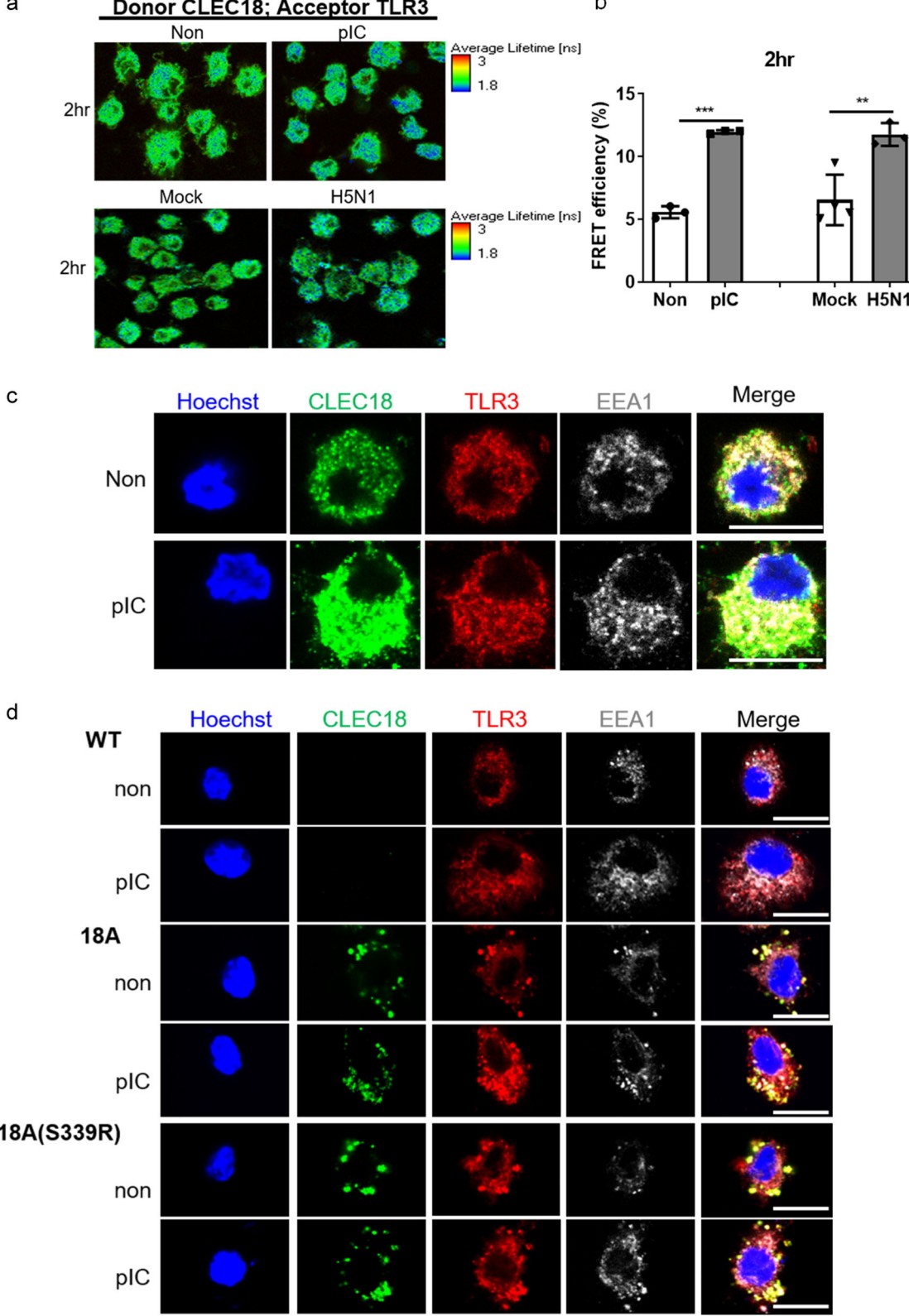

CLEC18A and ROSA-CLEC18A(S339R) mice after H5N1 IAV infection (Fig. 8c). Moreover, virus titer was suppressed dramatically in CLEC18A(S339R) mice (Fig. 8d), and the expression of viral M and NP mRNAs is downregulated in ROSA-CLEC18A and ROSA-CLEC18A(S339R) mice at day 7 post infection (Fig. 8e). We further investigate the cytokine amount in bronchoalveolar lavage fluid (BALF) by enzyme-linked immunosorbent assay (ELISA) (Fig. 8f). At days 4 and 7 post infection, higher levels of IFN-α, IFN-β, and IP-10 were noted in ROSA-CLEC18A and ROSA-CLEC18A(S339R) mice. Moreover, a higher level of IFNλ2/3 was observed on day 4, but not on day 7 post infection, while the expression of proinflammatory cytokines (TNF-α, CCL2, and CCL5) was similar in WT, ROSA-CLEC18A, and ROSA-CLEC18A(S339R) mice (Fig. 8f). These observations

**Fig. 4 CLEC18A and CLEC18A(S339R) associate with TLR3 and bind poly (I:C) directly. a, b** Human macrophages ($2 \times 10^5$) were incubated with HMW poly (I:C) or H5N1 virus (MOI = 2) for 2 h, followed by incubation with mouse anti-human CLEC18 mAb and rabbit anti-TLR3 mAb. After washing, samples were incubated with Alexa Fluor 488-conjugated goat anti-mouse IgG mAb (donor) and Alexa Fluor 546-conjugated goat anti-rabbit IgG mAb (acceptor). Cells were examined under a confocal microscope (TCS-SP5-MP-SMD, Leica), and images were analyzed by the "SymPhoTime" software. The image of the fluorescence lifetime is shown by false colors. Green color represents a long lifetime of CLEC18A, whereas blue blue color represents the reduced lifetime attributed to fluorescence resonance emission time (FRET) between Alexa Fluor 488 and Alexa Fluor 546 (n = 3–4). **c, d** Human macrophages ($6 \times 10^4$/well) (**c**) and BMDMs ($6 \times 10^4$/well) from WT and CLEC18 KI mice (**d**) were incubated with HMW poly (I:C) for 2 h, followed by immunostaining using mAb against CLEC18A/CLEC18A(S339R), TLR3, and EEA1, respectively. Cells were examined by a confocal microscope (TCS-SP5-MP-SMD, Leica). Green color: human CLEC18A and CLEC18A(S339R); red color: TLR3; white color: EEA1. pIC poly (I:C). Scale bar: 10 μM (n = 3). *P < 0.05, **P < 0.01 and ***P < 0.001.

indicated that CLEC18A and CLEC18A(S339R) are able to upregulate the expression of type I and type III IFNs to suppress H5N1 IAV replication in vivo.

## Discussion

It has been demonstrated that human macrophages and DCs only produce type I IFNs, but not TNF and IL-6, after TLR3 ligand stimulation. In contrast, mouse macrophages produce both type I IFNs and proinflammatory cytokine after poly (I:C) stimulation[4]. However, the underlying mechanism of the differential responses between human and mouse cell after poly (I:C) stimulation is still unknown. In this work, we demonstrated that human CLEC18A and CLEC18A(S339R) associate with TLR3, and act as co-receptor to bind poly (I:C). Interestingly, enhanced expression of type I and type III IFNs was noted in macrophages and epithelial cells isolated from ROSA-CLEC18A and CLEC18A(S339R) mice, respectively.

It is surprising to find that poly (I:C) binds to CLEC18A ($K_D$ = 2.10E − 08) and CLEC18A(S339R) ($K_D$ = 5.59E − 09) with high affinity. Both CLEC18A and CLEC18A(S339R) are typical C-type lectins with "QPD," "WND," and "WIGL" motifs. The presence of QPD motif is in accord with its preferential binding to galactan[1], while the "WND" motif is shown to promote binding to galactose and *N*-acetylgalactosomine[16]. Furthermore, the characteristic "WIGL" motif is highly conserved in canonical CTLD sequences[16]. Thus, CLEC18A is the first canonical C-type lectin that can bind poly (I:C) and acts as TLR3 co-receptor to enhance anti-viral immunity.

It is interesting to note that a single amino acid change [$S_{339} \rightarrow R_{339}$] in CTLD domain has a potent effect to enhance type I and type III IFN productions and anti-viral immunity. Compared to BMDM-CLEC18A, BMDM-CLEC18A(S339R) expressed higher levels of IFNs (Fig. 2b) and ISGs (Fig. 2c) with lower level of viral NP gene (Fig. 2d) expression after H5N1 IAV infection. Moreover, higher expression of IFN-β with a lower level of viral M gene was found in lung epithelial cells isolated from ROSA-CLEC18A(S339R) mice (Fig. 3b). This observation is in accord with a stronger induction of p-TBK1 and p-IRF3 in CLEC18A(S339R) (Fig. 5c). Furthermore, higher levels of type III IFN (Fig. 8f) with lower levels of virus titer and less lung inflammation were found in ROSA-CLEC18A(S339R) mice (Fig. 8c, d). The differential responses between CLEC18A and CLEC18A(S339R) alleles are in accord with a higher survival rate of ROSA-CLEC18A(S339R) mice after H5N1 IAV infection (Fig. 8b). Thus, the polymorphic amino acid residue S339R in the CTLD of CLEC18A seems contributing significantly in host immunity to H5N1 infection. Further study is needed to confirm the role of CLEC18A(S339R) allele in host defense against various microbial infections in the future.

In this work, we demonstrated that both CLEC18A and CLEC18A(S339R) bind to poly (I:C) and enhances poly (I:C)-induced IFNs production via forming a heterocomplex with TLR3. The higher affinity of CLEC18A(S339R) than CLEC18A may be attributed to the positive charge of $R_{339}$ located in the CLEC18A(S339R) CTLD, thereby increasing its binding to negatively charged poly (I:C). Moreover, ROSA-CLEC18A (S339R) mice produce higher amount of type I IFNs than ROSA-CLEC18A mice, and are more resistant to H5N1 IAV infection. These observations suggest that ROSA-CLEC18A(S339R) allele may be beneficial to host upon RNA virus infection. This speculation is supported by our observation that DV induced higher amount of IFNs in ROSA-CLEC18A(S339R) than ROSA-CLEC18A and WT mice. Among the species we examined[1], *Takifugu rubripes* CLEC18A contains N corresponding to $S/R_{339}$ in CTLD, while mice also contain "S" in CTLD domain. In addition to amino acid residue 339, another polymorphic amino acid residue was found at 421 in CTLD domain [($D_{421}$ (CLEC18A)→$N_{421}$ (CLEC18C)]. The $D_{421} \rightarrow N_{421}$ mutation converts the "WND" motif into WNN, and thus may attenuate their binding to galactose and *N*-acetylgalactosomine[16], and even other glycan and non-glycan ligands. In addition to CTLD domain, polymorphic amino acid residues located in N-terminal SCP/CAP domain was also noted[1]. Thus, CLEC18 is a highly polymorphic C-type lectin, and large-scale human population is necessary to understand whether another polymorphic amino acid residue exists in positions 339 and 421, and investigate the association of CLEC18A polymorphic amino acid residues in host defense against virus, bacteria, and other microbes in the future.

In previous studies, we found that human CLEC18 is detectable in culture supernatant[1] and human serum[17,18]. However, we cannot detect human CLEC18A/CLEC18A(S339R) in the ROSA-CLEC18 knock-in mouse serum. It may be attributed to lower expression of human CLEC18A/CLEC18A(S339R) in the knock-in mice, because the expression levels of CLEC18 in human immune cells and liver[1] are much higher than what we observe in ROSA-CLEC18A/CLEC18A(S339R) mice (Fig. 1a–c). Even though the functions of soluble CLEC18 are still unknown, CLEC18 serum level correlates with the stage of hepatitis B virus (HBV) infection and is a potential biomarker to predict HBeAg loss and seroconversion in CHB patients under nucleos(t)ide analog therapy[17]. Moreover, higher CLEC18 serum level positively correlated with viral loads[18] in HCV patients. These observations suggest that CLEC18 may also participate in the pathogenesis of liver inflammation caused by hepatitis B virus and HCV. Our observation that CLEC18A acts as a TLR3 co-receptor is the first step to explore its potential role in innate immunity against viral infection, and more studies are necessary to understand how CLEC18A contributes to host innate immunity against viral infections in the future.

## Methods

**Reagents.** Culture media and supplements were purchased from Gibco. Chemical reagents and chemical inhibitors were purchased from Sigma and Calbiochem, respectively. TLR ligands, including poly I:C (tlr-pic), cl097 (tlrl-c97-5), ODN2395 (tlrl-2395), Pam3CSK4 (tlrl-pms), and LPS (tlrl-3pelps) were purchased from InvivoGen. Murine or human macrophage colony-stimulating factor (M-CSF) and ELISA Kits (TNF, IL-6, IFN-λ2/3, CCL2, CCL3, CCL5, and IP-10) was obtained from R&D Systems. Anti-phospho-TBK1 (#5483), anti-TBK1 (#3504), anti-phospho-IRF3 (#4947), anti-IRF3 (#4302), anti-phospho-IκBα (#9246), and

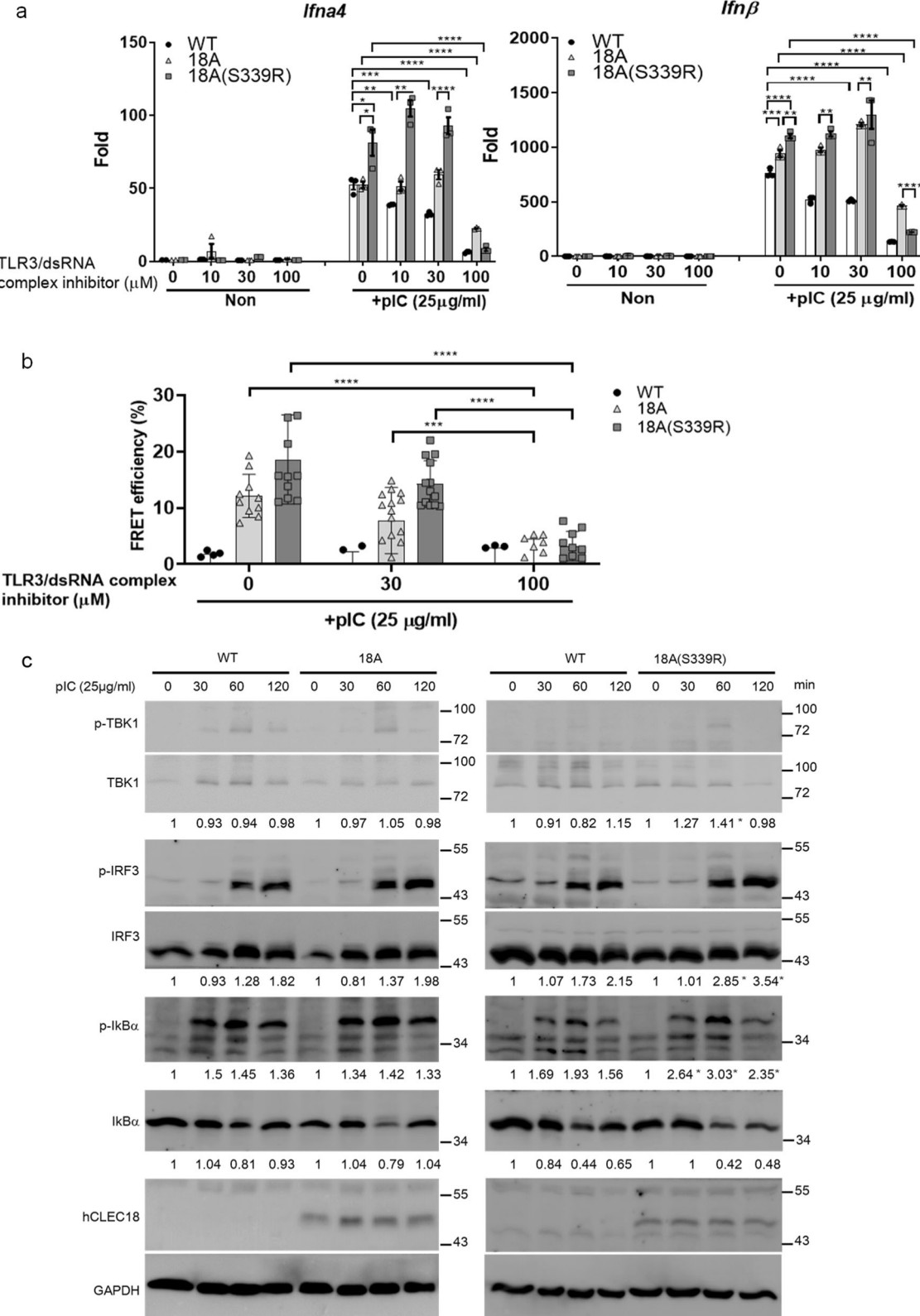

**Fig. 5 CLEC18A and CLEC18A(S339R) enhance TLR3-mediated signaling and IFN production. a** BMDMs (1 × 10^6/well) from WT and CLEC18 KI mice were preincubated with various concentrations of TLR3/dsRNA complex inhibitor for 1 h, followed by the addition of HMW poly (I:C) (25 μg/ml) for 12 h before harvesting samples to determine cytokine expression by RT-PCR (n = 3). **b** Alternatively, BMDMs (3 × 10^4/well) were preincubated with various concentrations of TLR3/dsRNA complex inhibitor for 1 h, followed by the addition of HMW poly (I:C) (25 μg/ml) for 2 h for FRET assay (n = 3-8).
**c** BMDMs (2.5 × 10^6/well) were incubated with HMW poly (I:C) and then lysates were harvested at indicated time points, followed by harvesting samples to analyze protein phosphorylation by WB. pIC poly (I:C), min minute (n = 3). *P < 0.05, **P < 0.01, ***P < 0.001 and ****P < 0.0001.

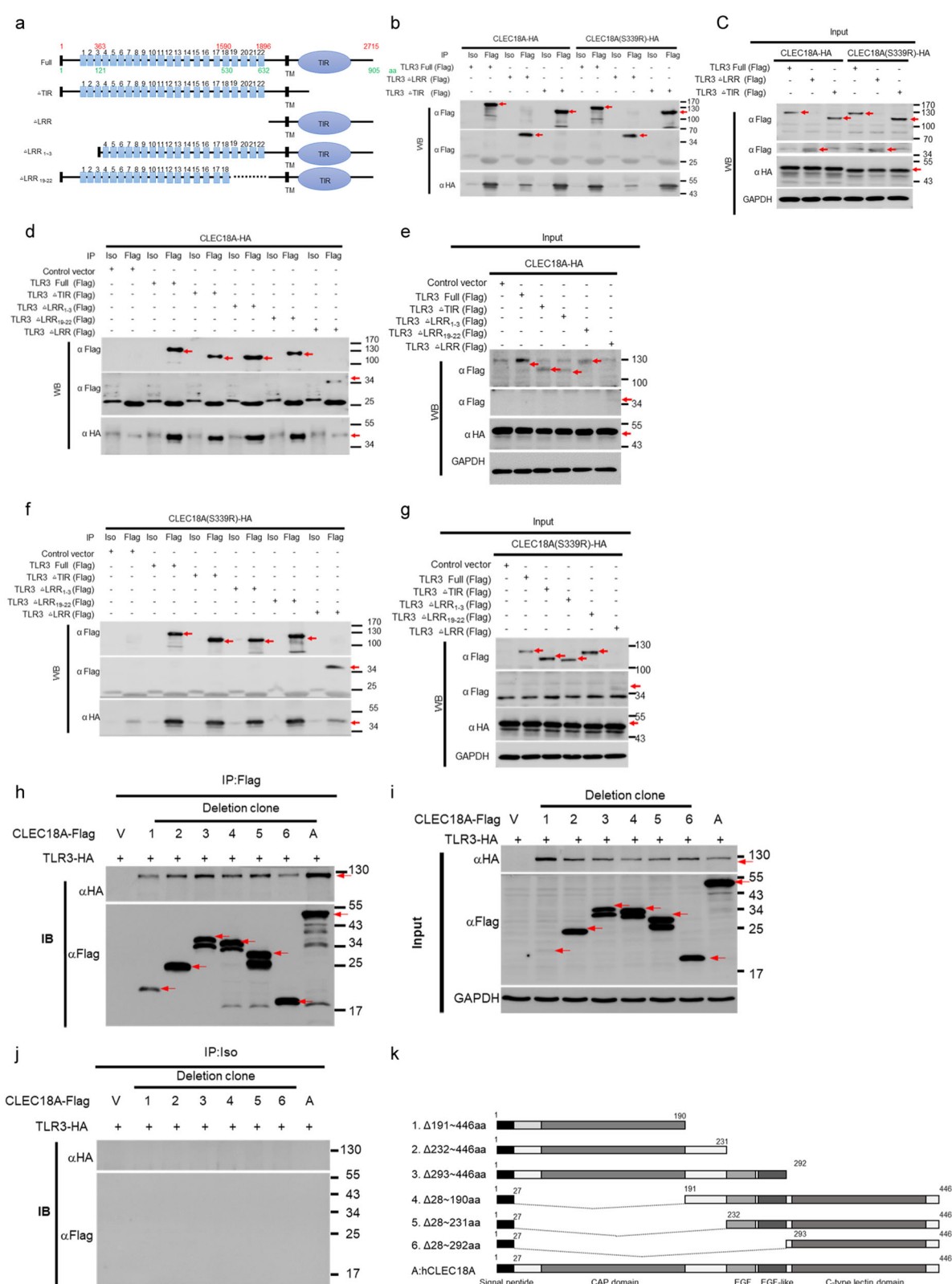

anti-IκBα (#4814) Abs were purchased from Cell Signaling Technology. TLR3 antibody (#GTX20260) and TLR7 antibody (#NBP2-24906) were purchased from Genetex and Novus Biologicals. TLR3/dsRNA complex inhibitor (#614310) was purchased from Calbiochem Merck Millipore. The transfection reagents used were as follows: ON-TARGETplus Non-targeting Control Pool (Dharmacon, D-001810-10-50), ON-TARGETplus Human TLR3 siRNA-SMARTpool (Dharmacon, L-007745-00-0020), HiPerFect Transfection Reagent (Qiagen, 301707), human CLEC18 siRNA#1 (Qiagen, SI03231088), human CLEC18 siRNA#2 (Qiagen,

SI04149502), AllStars Negative Control siRNA (Qiagen, 1027280), and AllStars Neg.siRNA AF 488 (Qiagen, 1027289).

**Mice**. Eight to twelve-week-old WT C57BL/6 and ROSA-CLEC18 mice were used for all the experiments. The ROSA-CLEC18A and ROSA-CLEC18A(S339R) knock-in mice were generated by the Transgenic Mouse Models Core. All animals were bred and housed at the Academia Sinica SPF animal facility and the

**Fig. 6 Mapping of TLR3 and CLEC18A/CLEC18A(S339R) interaction domain. a** Scheme of human TLR3 deletion mutants. **b–g** The 293T cells ($3 \times 10^6$) were transfected with pMACS-Kk-HA-hCLEC18A/CLEC18A(S339R) (7.5 μg) and pFlag-CMV1-hTLR3 (full-length and deletion mutants) (7.5 μg) for 48 h, followed by immunoprecipitation using anti-Flag mAb. Immunoprecipitates were fractionated on SDS-PAGE and blotted to PVDF membranes before probing with anti-HA mAb and anti-Flag mAb, respectively. Blots were further incubated with peroxidase-conjugated anti-mouse IgG antibody and developed by ECL Kit. **h–k** The 293T cells ($3 \times 10^6$) were transfected with pCMV-Tag4A-hCLEC18A (full-length and deletion mutants) and pUNO1-TLR3-HA for 48 h, followed by immunoprecipitation using anti-Flag mAb. Immunoprecipitates were fractionated on SDS-PAGE before blotting to PVDF membranes. Blots were probed with anti-HA mAb and anti-Flag mAb, respectively, followed by incubation with peroxidase-conjugate anti-mouse IgG antibody and developed by ECL Kit. Iso isotype control antibody, IP immunoprecipitation, WB western blot. Red arrows: CLEC18 or TLR3; V: vector only; [+]: Flag-tagged CLEC18A and deletion mutants were co-transfected with HA-tagged TLR3 expression vector (TLR3-HA). All WB experiments have been repeated at least three times.

Laboratory Animal Center of National Defense Medical Center. Animal experiments were approved by the Institutional Animal Care and Use Committee at AS core (protocol ID 18-12-1243), and in accordance with the recommendations in the Guide for the Care and Use of Laboratory Animals of the Taiwanese Council of Agriculture.

**Preparation of macrophages.** To prepare mouse macrophages, bone marrow cells were isolated from femurs and tibias and incubated in RPMI-1640 complete medium supplemented with 10% (v/v) fetal bovine serum and 10 ng/ml of recombinant mouse M-CSF (R&D Systems) for 6 days. To prepare human macrophages, CD14[+] monocytes isolated from peripheral blood mononuclear cells were incubated in RPMI-1640 medium supplemented with 10 ng/ml human M-CSF (R&D Systems) for 6 days to differentiate into macrophages[19]. The protocol of peripheral blood mononuclear cell isolation was approved by the Human Subject Research Ethics, Academia Sinica (Taiwan, number AS-IRB02-108175).

**Isolation of primary mouse alveolar epithelial type II cells.** C57BL/6 mice were sacrificed by an overdose of isoflurane. The inferior vena cava was cut and lungs were perfused with PBS via the right ventricle until it turned white in color. A small nick was made into the exposed trachea to insert a shortened 20 G angiocatheter that was fixed and a total volume of 3 ml of sterile dispase was injected (BD Biosciences), followed by injection of 500 μl of sterile 1% low-melting agarose in PBS (Lonza). After a few minutes of incubation, the lungs were removed and placed into a tube containing 1 ml of dispase for 45 min (25 °C). Lungs were then transferred into a culture dish containing Dulbecco's modified Eagle's medium (DMEM)/10 mM HEPES/100 U/ml DNaseI (Sigma)/antibiotics for 5 min, and the tissue was carefully dissected from the airways and large vessels. The single cell was successively suspended through a 100 and 40 μm strainers. The cell was centrifuged at $300 \times g$ for 10 min at 4 °C, then the cell was resuspended with RBC lysis buffer for 5 min to lyse the presence of RBCs, and finally, it was directly added to 10 ml DMEM supplemented with 10% FCS/HEPES and antibiotics. The cell was centrifuged and the resuspended cell was placed on prewashed 100-mm tissue culture plates that had been coated for 24–48 h at 4 °C with 42 μg CD45 and 16 μg CD32 (BD Pharmingen). After incubation for 2 h at 37 °C, type II cells were gently collected from the plate and filtered by 40 μm strainers to suspend the single cell. AECs were plated into 6-well cell culture plates at $1.5 \times 10^6$/well, followed by detecting prosurfactant protein C (Abcam, ab40879), the specific marker for alveolar epithelial type II cells, by flow cytometry to determine the purity.

**Transfection of macrophages and HT1080 cells by CLEC18 siRNA and CLEC18 expression vectors.** Human macrophages were seeded at $7 \times 10^4$ cells/well in 96-well microtitration plates. After incubation at 37 °C overnight, cells were transfected with 100 nM siRNA in 2.5 μl HiPerFect, and further incubated at 37 °C for 48 h before harvesting for the functional assay. Alternatively, HT1080 cells were seeded in 12-well microplate ($1.2 \times 10^5$ cells/well) and incubated at 37 °C overnight, followed by incubation with 10 nM siRNA in 6 μl HiPerFect at 37 °C for 48 h to knock down endogenous CLEC18. The siRNA-transfected HT1080 cell cells were then seeded in 12-well microplate ($1.2 \times 10^5$ cells/well) and incubated at 37 °C overnight, followed by transfection with 1 μg plasmids in 4 μl Turbofect transfection reagent (Thermo Scientific™, catalog number R0532), and further incubated at 37 °C for 24 h before functional assay.

**Detection of human and mice CLEC18 by Western blot or flow cytometry.** For Western blotting analysis, cells ($1 \times 10^6$) were lysed by RIPA buffer, followed by fractionation on 12% SDS-PAGE before blotting onto PVDF membrane. Membranes were probed with anti-hCLEC18 mAb (clone 3A9E6 (1)) or anti-mCLEC18 mAb (home-made antibody C8G), followed by incubation with peroxidase-conjugated goat anti-mouse polyclonal antibody (Millipore, AP181P) or donkey anti-human polyclonal antibody (Jackson ImmunoResearch, 709-035-149). Blots were developed by the Immobilon™ Western Chemiluminescent using horseradish peroxidase as the substrate (Millipore™). For flow cytometry analysis, primary cells were incubated with phycoerythrin (PE)-conjugated anti-mouse CD4 antibody (BioLegend, 100408), PerCP-Cy5.5-conjugated rat anti-mouse CD8a (BD,

561109), PE/cyanine7-conjugated anti-mouse CD11b antibody (BioLegend, 101216), Alexa Fluor 488-conjugated anti-mouse Ly6G/Ly6C (Gr-1) antibody (BioLegend, 108417), Brilliant Violet 510-conjugated anti-mouse IA/IE antibody (BioLegend, 107636), Brilliant Violet 421-conjugated anti-mouse/human CD45R/B220 antibody (BioLegend, 103251), and then intracellular stain with Alexa Fluor 647-conjugated anti-mCLEC18 mAb (home-made antibody clone C5F). After staining, cells were analyzed by a FACSVerse™ flow cytometer (BD Biosciences) and data were analyzed using the FlowJo software.

**Virus.** Original stocks of recombinant PR8 expressing HA and NA derived from either VN1203 (VNHA,NA) or 127 HK5498 (HKHA,NA) were obtained from HKU-Pasteur Research Pole[20,21]. Both IAV strains were grown in Madin–Darby canine kidney cells (MDCK, ATCC®CCL-34™) according to a standard procedure and virus titer was determined by a plaque assay on confluent MDCK cells[22]. DVs (DV2-PL046) were propagated in C6/36 mosquito cell line, and viral titers were determined by plaque-forming assays using BHK-21 cells.

**Inoculation of virus.** Murine and human macrophages were infected with IAVs at a multiplicity of infection (MOI) = 2. After 1 h adsorption, cells were washed once with PBS, followed by incubation with culture medium (RPMI supplemented with 10% fetal bovine serum) for 12 h or 24 h. WT or ROSA-CLEC18 knock-in mice (9–12 weeks old) were anesthetized by intraperitoneal injection of ketamine and xylazine, followed by intranasal inoculation with H5N1[(HA,NA)] virus[20] ($1.5 \times 10^3$ PFU) in 20 μl sterile PBS. Survival rate and body weight loss were monitored until day 21 post infection.

**Immunofluorescence staining and fluorescence lifetime imaging microscopy-FRET.** Murine and human macrophages were fixed and permeabilized, followed by incubation with anti-CLEC18 (60 μg/ml), anti-TLR3 (20 μg/ml), or anti-TLR7 (20 μg/ml) Abs at 4 °C for 16–18 h, followed by incubation with Alexa Fluor 488-conjugated donkey anti-mouse IgG (Jackson ImmunoResearch, 715-545-150) or Alexa Fluor 546-conjugated goat anti-rabbit IgG (Life Technologies, #A-11035). Samples were mounted with coverslips and observed under a confocal microscope (TCS-SP5-MP-SMD, Leica) and analyzed by the SymPhoTime software.

**Immunoprecipitation and Western blotting.** To understand whether hCLEC18A or hCLEC18A(S339R) associates with hTLR3, pMACS.Kk-HA(C), pMACS.Kk-HA (C)-hCLEC18A, and pMACS.Kk-HA(C)-hCLEC18A(S339R) (7.5 μg) were co-transfected with pFlag-CMV1-hTLR3 (7.5 μg), respectively, into 293T cells ($3 \times 10^6$), followed by incubation at 37 °C for 48 h before harvesting. Cell lysates were preincubated with Sepharose beads to remove non-specific associated protein, followed by incubation with anti-Flag mAb-conjugated Sepharose beads (Sigma-Aldrich, A2220) or isotype mAb-conjugated Sepharose beads to pull down TLR3-associated proteins. The immunoprecipitates were fractionated on 12% SDS-PAGE and blotted onto PVDF membranes, which were probed with anti-Flag mAb and anti-HA mAb before incubation with peroxidase-conjugated goat anti-mouse polyclonal antibody (Millipore, AP181P or AP200P). Blots were visualized by adding Immobilon™ western chemiluminescent horseradish peroxidase substrate (Millipore™). To determine CLEC18 domains associated with TLR3, pUNO1-hTLR3 (7.5 μg) and pCMV-Tag4A-hCLEC18A (7.5 μg) deletion mutants were transfected into 293T cells ($3 \times 10^6$), followed by incubation at 37 °C for 48 h before immunoprecipitation.

**RNA extraction and real-time PCR.** Total RNA was extracted by TRIzol reagent (Invitrogen™) according to the vendor's instruction. First-strand cDNA was synthesized using a RevertAid First-Strand cDNA Synthesis Kit (Thermo Scientific). PCR reaction was performed by LightCycler® 480 System (Roche Applied Science). All primers are listed in Supplementary Table I.

**Quantification of viral M and NP gene.** To determine M and NP expression in infected mice, samples were harvested from infected mice for RNA extraction

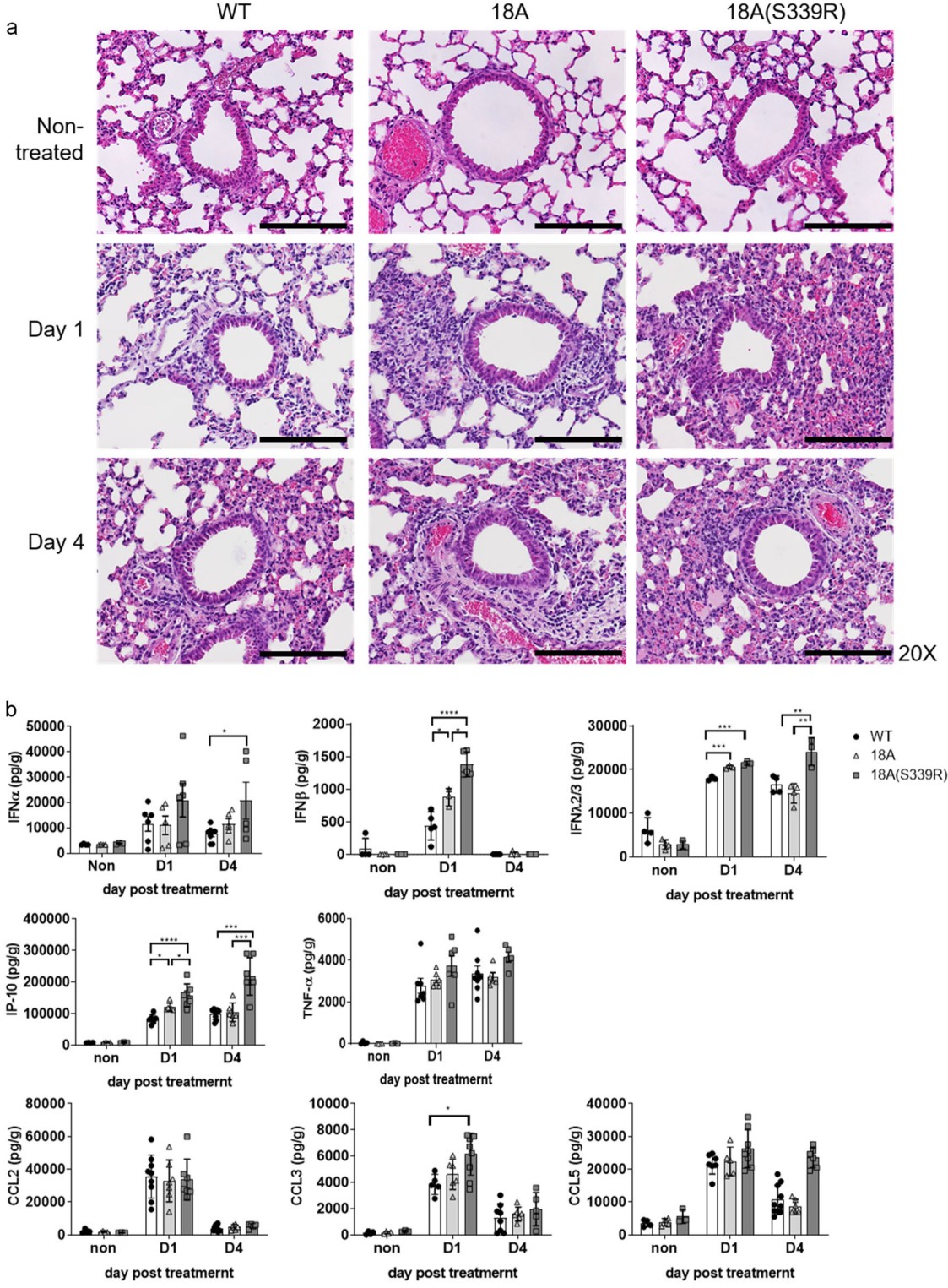

**Fig. 7 CLEC18 enhances poly (I:C)-induced inflammatory reaction in vivo.** WT, ROSA-CLEC18A, and ROSA-CLEC18A(S339R) mice were inoculated with HMW poly (I:C) (30 μg per mouse) via intranasal route, and then were sacrificed at day 1 (D1) and 4 (D4) post treatment. Lung sections were subjected to hematoxylin and eosin (H&E) staining (*n* = 3–6) (**a**), and cytokine in bronchoalveolar lavage was determined by ELISA (*n* = 3–8) (**b**). pIC: poly (I:C). One-way ANOVA was performed. *P < 0.05, **P < 0.01, ***P < 0.001 and ****P < 0.0001 for WT group versus groups of CLEC18A or CLEC18A(S339R). Scale bar: 200 μM.

using the TriRNA Pure Kit (Geneaid, Catalog No.TRP100), and first-strand cDNA was synthesized using a RevertAid First-Strand cDNA Synthesis Kit (Thermo Scientific) as per the vendor's instruction. Quantitative real-time PCR analysis was performed using the LightCycler® 480 System (Roche Applied Science).

**Determination of virus titers in lung tissue**. Lungs isolated from IAV-infected mice were homogenized with a homogenizer (MagNA Lyser Instrument, Roche Applied Science) in 0.7 ml PBS on ice, followed by centrifugation at 11,000 r.p.m. for 10 min at 4 °C. Supernatants were diluted serially in DMEM, and virus titers were determined by plaque assay on confluent MDCK cells.

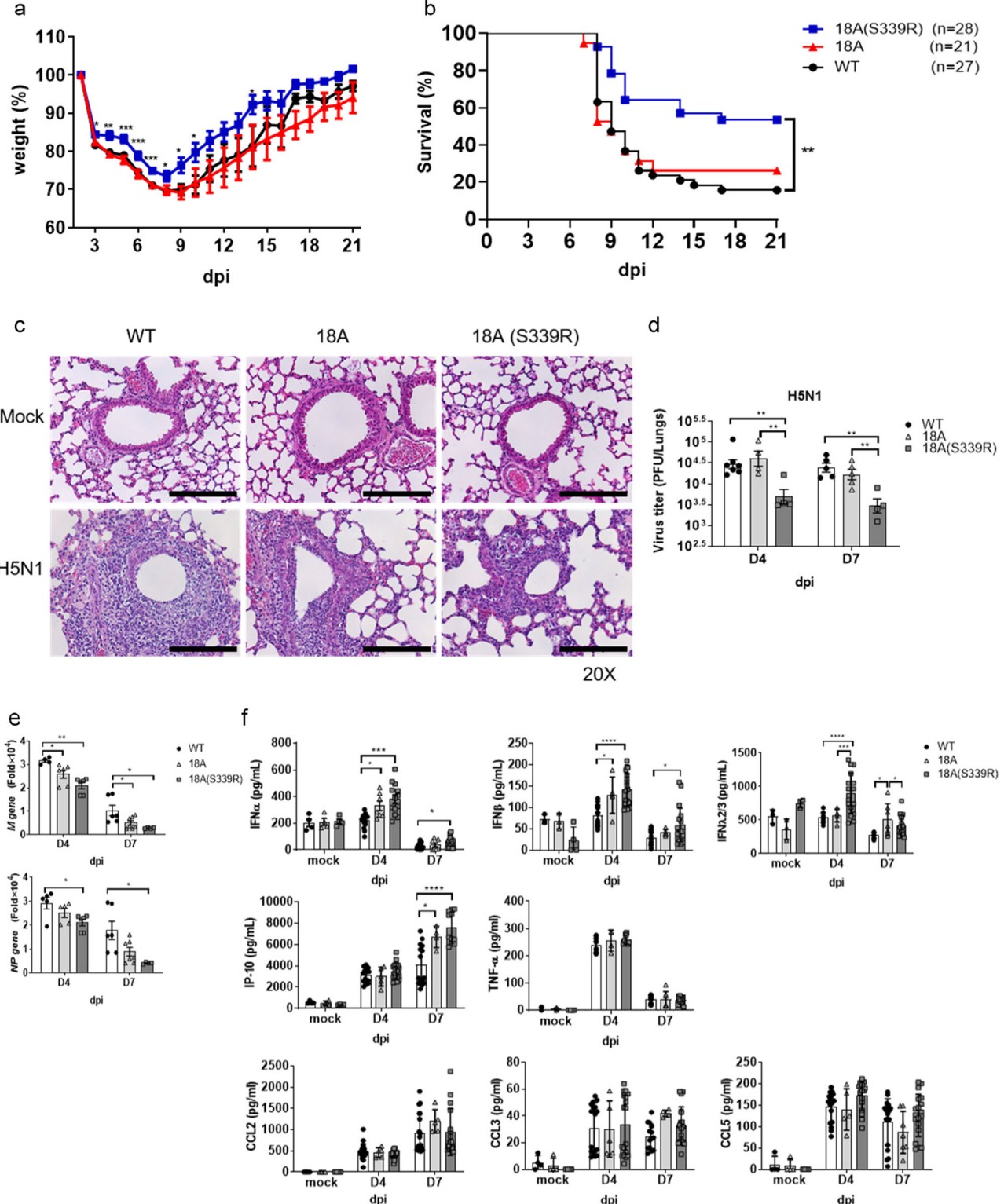

**Fig. 8 CLEC18A and CLEC18A(S339R) protect host from IAV-induced inflammation and lethality. a**, **b** WT, ROSA-CLEC18A, and ROSA-CLEC18A (S339R) mice were inoculated with H5N1 virus ($1.5 \times 10^3$ PFU) by intranasal route to observe body weight changes (**a**) and survival rate (**b**) ($n = 21$–28). **c** Lung tissues harvested at day 4 post infection were subjected to hematoxylin and eosin staining ($n = 3$). **d**, **e** Virus titer in lung tissues was determined by plaque assay ($n = 4$–6) (**d**), while viral structure gene expression (M and NP) was determined by RT-PCR ($n = 4$–6) (**e**). **f** Cytokines contained in the bronchial alveolar lavage fluid (BALF) from WT and ROSA-CLEC18A/CLEC18A(S339R) mice at days 4 and 7 post infection were determined by ELISA ($n = 3$–17). d.p.i. days post infection. One-way ANOVA was performed. $*P < 0.05$, $**P < 0.01$, $***P < 0.001$ and $****P < 0.0001$ for WT group versus group of CLEC18A or CLEC18A(S339R). Scale bar: 200 µM.

**Determination of cytokine levels by ELISA**. The levels of cytokines in culture supernatant and BALF were determined by ELISA Kits according to the vendor's instruction. All samples were stored at −80 °C before assay.

**Administration of poly I:C to mice by intranasal route**. Mice were anesthetized by intraperitoneal injection of ketamine and xylazine, followed by intranasal administration of saline or 30 μg HMW poly I:C (InvivoGen) in 20 μl sterile saline. Poly I:C was freshly prepared for each experiment according to the vendor's instruction.

**Bronchoalveolar lavage**. All the cell infiltrates and contents in lung tissues were harvested by infusing 3 ml PBS to lung six times. All the samples were centrifuged at 1600 r.p.m. for 10 min at 4 °C and the supernatant was stored at −80 °C before use.

**Statistics and reproducibility**. Data were analysed using GraphPad Prism 8 or FlowJo, and the data were shown as mean ± SEM. Statistical significance was determined using unpaired one- or two-way analysis of variance with the Tukey's multiple comparison test. The statistical significance of survival rate was determined using log-rank test. *$P < 0.05$, **$P < 0.01$, and ***$P < 0.001$

**Reporting summary**. Further information on research design is available in the Nature Research Reporting Summary linked to this article.

## Data availability

All data generated or analyzed during this study are included in this published article (and its Supplementary information files). The source data underlying graphs shown in the main figures are presented in Supplementary Data 1. Full blots are shown in Supplementary Fig. 5.

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

## Acknowledgements

This work was supported by the Academia Sinica Investigator Award (AS-IA-109-L02), Summit Research Projects and grant 109-2101-01-19-20 (Academia Sinica), Ministry of Science and Technology (MOST 107-2321-B-001-015), and VGH, TSGH, and AS Joint Research Program (VTA109-A-3-1). We thank the Taiwan Mouse Clinic (TMC) and Transgenic Mouse Model Core Facility (TMMC), which are funded by the National Research Program for Genomic Medicine (NRPGM) at the National Science Council of Taiwan, for technical support in histopathology experiments.

## Author contributions

Y.-L.H: designed, performed experiments, data analysis, and wrote the manuscript; M.-T.H., and P.-S.S.: provided technical support; T.-Y.C.: provided technical support; R.-B.Y.: provided reagents; A.-S.Y. and C.-M.Y.: generation of anti-mouse CLEC18A mAb; W.-C.C. and Y.-W.H.: provided technical support; S.-L.H: designed, performed experiments, data analysis, and wrote the manuscript

## Competing interests

The authors declare no competing interests.
