## [Peer Review File · Communications Biology]

Reviewers' comments:

Reviewer #1 (Remarks to the Author):

The authors present a nice study demonstrating the involvement of C-type lectin, CLEC18A in TLR3 signalling. Interestingly the expression pattern of this accessory molecule appears to be similar to others reported for TLR3 i.e. TRIL (predominantly in the brain). The study demonstrates the interaction of CLEC18A with TLR3 and poly I:C through a variety of techniques. The authors take this further by identifying a Serine residue that lies in the C-Type lectin domain, CTLD of CLEC18A important for the binding of CLEC18 to TLR3. The study also demonstrates the importance of CLEC18A and the serine mutant form CLEC18A-1 in the induction of IFNs following viral infection (H5N1 IAV and Dengue Virus) whereby they decreases viral titres, viral protein levels and mortality in vivo. Collectively, the study highlights the importance of this accessory molecule and specifically the serine residue within the C-type lectin domain of CLEC18A for adequate IFN responses following TLR3 engagement most likely in the brain. This work would be of significant interest to researchers in the field with limited publications available on this molecule. Some major and minor comments that may improve the manuscript prior to publication are detailed below.

Major

1. Line 233- 235: Authors state that unlike IFN α and IFN β , ROSA-CLEC18A and ROSA-CLEC18A-1 BMDMs do not upregulate TNF- α and IL-6 in Fig. 2a. However, TNF and IL-6 are both clearly upregulated on the RNA (Fig. 2a) and protein level (Supp Fig. 1a) the difference is simply that expression is not significantly increased compared to WT BMDM levels. This is important to note and suggests that the mouse model in which human CLEC18A is overexpressed may not match what is known to be true for TLR3 responses in human macrophage. Importantly however, this does not negate the significant amount of data in the remainder of the paper demonstrating the interaction of CLEC18A/A-1 with poly I:C and TLR3.
2. It is interesting that in this system CLEC18A-1 seems to be inhibitory on the TRAF6/NF- κ B side of the TLR3 pathway following polyI:C stimulation (Supp Fig 1a), yet enhances the TRIF side. On the other hand in CLEC18A BMDM, 'endogenous' CLEC18A appears important for both. Have the authors identified the residue responsible for the differences we observe between human and murine signalling in response to poly I:C?? What's the conservation in sequence between human and mouse. Data in supplemental Fig1a only appears to have one data point. Can authors please clarify if this is the mean of technical triplicate, how many experiments or perform additional experiments to satisfy n number requirements for the conclusions drawn.
3. The TLR3/dsRNA inhibitor suppresses IFN β at the same concentration in WT and CLEC18 and CLEC18A-1 cells (100uM) (Fig. 4c). This is also the case for IFN α . This result weakens the claim that C18A is infact a co-receptor for TLR3. Could the TLR3/dsRNA inhibitor be added to FRET experiments to definitively prove C18A/C18A-1 as co-receptors of TLR3? If not perhaps remove. The bio-layer interferometry data is much more convincing.
4. The confocal data in supplemental showing EEA1, TLR3 and CLEC18 co-localisation is really nice and could perhaps be moved to the main figures.
5. Figure 5b – it is unfortunate that the authors present the input blots for the immunoprecipitation studies on separate gels/western blots. It is generally expected that the Input/expression of plasmids in both the Isotype control lane and the Flag lanes be shown to confirm adequate expression of plasmid in these samples and in order to confirm a legitimate interaction. Blots in Figure 5d with deletion clones shows this appropriately.
6. Stats are needed in figure 7 for survival. Number of mice not indicated in text or figure legend. This should definitively be included prior to publication.

Minor

1. The labelling of human Macrophages as M-M in figure legends is very confusing for the reader. Prior to reading the legend the reader may think this data represents values from Mouse Macrophages (M-M). Can authors please explain or change this to H-M or other?
2. Figure 1, Expression in mouse DC is not as convincing as in BMDM however, I appreciate the antibodies may not be great and the majority of the data in the paper is in BMDM.

Reviewer #2 (Remarks to the Author):

The manuscript by Ya-Lang Huang et al. roles of human CLEC18A function in TLR3 dependent signaling event. They generated transgenic mice expressing human CLEC18A and mutant CLEC18A S339R for ex vivo and in vivo experiment. Author firstly demonstrated Ifn and other cytokines expression in BMDM and alveolar epithelial cells and found that hCLEC18A and mutant CLEC18A expression enhances expression of Ifnb and after poly IC stimulation and IAV and DV infection. Then author demonstrated TLR3 and CLEC18a associated each other. Author performed in vivo inflammatory model using poly I:C and found that CLEC18A and mutant CLEC18A mice showed severe inflammation. In contrary, IAV infection to CLEC18A and mutant CLEC18A mice showed reduced inflammatory response and prolonged survival comparing to wild mice.

Author tried to explain the discrepancy of TLR3 innate immune response between mice and human, and CLEC18A may cause changes of TLR3 response in human and mice. However, logics of manuscript and these results are not fully support the author's point. In addition, control of experiments in each experiment and references are unsuitable.

The concept is certainly intriguing and is of interest. However, there are several limitations in the data as following.

1. In the introduction, explanation of mutant CLEC18A-1 is unclear. Author should what are causes by the mutation in CLEC18A function. Mutant for CLEC18A-1 is should show like "CLEC18A S339R".
2. Ref 6 is not suitable. Ref 6 did not include the results of infection by H3N2 IAV.
3. Explanation of Ref 8 is incorrect. Ref 8 is results of human cells.
4. Author mentioned that Clec18A is low expression in BMDM, however expression level of endogenous clec18a is unclear in Fig1.
5. In Fig2a and b, author included results of human macrophages. These are completely different cell line from mouse cells, therefore it is not suitable to put in the same panel and cannot discuss the production level of Ifns in between mouse and human by these results.
6. Poly I:C treatment in vivo showed increase of inflammation in CLEC18A and mutant CLEC18A mice compared with wild type mice, however IAV treatment in these mice showed reduces of inflammation. Why these difference induced by these stimulation?
7. It is unclear why author use CLEC18A mutant mice to show difference of TLR3 responses between human and mice.
8. Author tried to explain the discrepancy of TLR3 innate immune response between mice and human using transgenic mice which expressed human CLEC18A. However, ectopic expression in mice to human gene does not help to explain the difference of human and mice. It is hardly compared with wild type and human gene transgenic mice. Author should compare mice gene and human gene transgenic mice. If author say the expression of Clec18a in mice is low, author show the evidence of expression level in human and mice and use knockdown or knockout technique in human cells line.

Overall, the data shown do not support the main conclusions drawn by the authors.

Reviewer #3 (Remarks to the Author):

In this study, Huang et al. described essential roles of human CLEC18A and CLEC18A-1 in TLR3 signaling pathway by utilizing the ROSA-CLEC18A and ROSA-CLEC18A-1 mice. Their results suggested that human CLEC18A and CLEC18A-1 directly interacted with TLR3 and poly(I:C), enhancing IFN production without the enhancement of pro-inflammatory cytokine production. ROSA-CLEC18A and ROSA-CLEC18A-1 mice were more resistant to H5N1 IAV infection compared to WT mice, suggesting that these proteins are important in antiviral immune responses in human. Overall, their findings that human CLEC18A and CLEC18A-1 act as a co-receptor with TLR3 is particularly interesting in view of differential roles of them between human and mouse, in TLR3-mediated antiviral responses. However, data presented in this manuscript is not sufficient to support their conclusion. Furthermore, there seems to be over-interpretation throughout their manuscript. To strengthen their conclusion, many concerns listed below should be addressed.

Specific comments

1. Figure S1A, S2B and S2C should be explained in their manuscript.
2. In figure 2A, data showing IP-10 production in response to poly(I:C) (1 microG/mL) is missing.
3. In figure 2A and S1A, there seems to be some discrepancy, especially data on pro-inflammatory cytokines, they should kindly explain about that. Furthermore, they should place ELISA data but not qPCR data onto the main figure.
4. Figure 2a did not correspond to supplementary fig. 1a. While mRNA level of TNF α showed no significant difference between WT and CLEC18 expressing cells, the protein level was increased in CLEC18 expressing cells.
5. They should describe about how they prepared human macrophages in [materials and methods] section.
6. In figure 4D, to support their conclusion, they should perform statistical analysis.
7. To strengthen their conclusion, they should perform knockdown experiment using human cells.
8. In figure 6 and 7, they should show data using control (untreated) mice.
9. In figure 1A, because the expression level of mRNA does not always reflect that of protein, they should show CLEC18 protein expression in each tissues and cells (especially immune cells). Same for figure 1B.
10. Because their previous paper (ref. 1) also suggests that CLEC18 is secreted by cells, they should clearly refer it in their manuscript.
11. In figure 1C, they should clearly describe about the antibody they used in figure legend. Furthermore, in case that they used antibody against endogenous human CLEC18 protein but not against tags, they should clearly describe about the information of the antibody used.
12. There are many typos and grammatical errors throughout the manuscript that should be corrected (e.g. [purchased] at line 101, [C57BL/J] at line 118 and [analysis] at line 133 should be [were purchased], [C57BL/6] and [analysis of], respectively.).
13. At line 133 and 134, they should not use the abbreviation [AEC] and [ATII] without prior explanation.
14. Although they claim that CLEC18A-1 is more potent to induce interferon response, this reviewer could not find any statistical analysis between CLEC18A and CLEC18A-1.
15. In figure 2A and 2B, they claim that CLEC18A and CLEC18A-1 increased type I interferon expression when even 1.1-1.2 fold. On the other hand, they claim that, in figure S1B, S1C and S1D, CLEC18 did not affect to ifna or ifnb mRNA expression even 2-4 fold. This is quite strange. They should perform statistical analysis in figure S1 and precisely interpret their data.
16. In figure 4C, statistical analysis seems to be strange. The difference in ifnb between WT (pIC 0) and WT (pIC 30) was significant, but that between WT (pIC 0) and WT (pIC 10) was not significant. If

the figure is correct, latter also seems to be significant. They should confirm the method of statistical analysis throughout their manuscript.

17. In figure 5D, the means of [+] and [-] is not clear.

18. In figure 5D, it is interesting to investigate which domain of CLEC18 is involved in its binding to poly(I:C).

Reviewers' comments:

Reviewer #1 (Remarks to the Author):

Major

1. Line 233-235: Authors state that unlike IFN α and IFN β , ROSA-CLEC18A and ROSA-CLEC18A-1 BMDMs do not upregulate TNF- α and IL-6 in Fig. 2a. However, TNF and IL-6 are both clearly upregulated on the RNA (Fig. 2a) and protein level (Supp Fig. 1a) the difference is simply that expression is not significantly increased compared to WT BMDM levels. This is important to note and suggests that the mouse model in which human CLEC18A is overexpressed may not match what is known to be true for TLR3 responses in human macrophage. Importantly however, this does not negate the significant amount of data in the remainder of the paper demonstrating the interaction of CLEC18A/A-1 with poly I:C and TLR3.

Answer:

- 1) *We agree with reviewer's comment and change the description accordingly. The original Fig. 2a is mRNA level, and we change in to protein level in this revised version. In addition, we also repeat this experiment with two different doses of poly (I:C) to show dose-dependent effect. We also compared the effect of poly (I:C) with other TLR ligand in cytokine expression in mRNA and protein levels (Supplementary Fig. 2) in this revised version.*
- 2) *To confirm the effect of CLEC18 in human macrophages, we further knock down endogenous CLEC18 in human macrophages (Supplementary Fig. 3). We found that CLEC18 knockdown in human cells only affects poly (I:C) -, but not LPS-, induced cytokine production (Supplementary Fig. 3). This observation further supports the argument that CLEC18 is the co-receptor of TLR3.*

2. It is interesting that in this system CLEC18A-1 seems to be inhibitory on the TRAF6/NF- κ B side of the TLR3 pathway following poly (I:C) stimulation (Supp Fig 1a), yet enhances the TRIF side. On the other hands in CLEC18A BMDM, 'endogenous' CLEC18A appears important for both. Have the authors identified the residue responsible for the differences we observe between human and murine signaling in response to poly I:C? What's the conservation in sequence between human and mouse

Answer:

- 1) *As we show in Figure 1 a-c, mouse leukocytes do not express endogenous CLEC18A, neither can we detect CLEC18A in mouse lung tissue. Thus, the cytokine secretion in wild type mouse immune cells and lung tissue is only via TLR3, without the involvement of mouse endogenous CLEC18A.*
- 2) *The pairwise alignment of human and mouse CLEC18 is as shown in the following page.*
- 3) *The identity and consensus position of human and mice CLEC18A CTLD is 91.2% and 94.2%, respectively. The corresponding amino acid at position 339 is S in both human and mouse CLEC18A. We do not find R339 in available mouse CLEC18 sequence in database.*

Human CLEC18A ₃₃₁GGVLAQIKSQKVQDILAFYL₃₄₀

Mouse CLEC18A ₃₃₁GGVLAQIESQKVQDILAFYL₃₄₀

Red box: CTLD region

	(1)	10	20	30	40	50	60	70	81	Section 1		
NP_853527-mCLEC18A	(1)	MIQPEPSLWIGSQSRWLG	LLLLLLSLLGLTWTVEVQ	PPQ-PKQDPTLQALS	SRKESFLILTAHNR	LSRVHPPAANM	ORDWS					
NP_872425-hCLEC18A	(1)	MLHPETSPGRG-----	HLLAVLLALLGTAWAEV	WFPPOLOEQAPMAGAL	NRKESFLLLSLHN	RLRSWVOPPA	DMRRLDWS					
Consensus	(1)	MIPE S G	LL LLLALLG	W EV PPO	Q P	AL RKESFLILS	HNRLRS V	FPAA M	RLDWS			
	(82)	82	90	100	110	120	130	140	150	162	Section 2	
NP_853527-mCLEC18A	(81)	ESLAQLAEARAALCVTSV	TNLPASTPGHNSHVGNV	QVLMFMGSA SFVEV	VNLFNFAEGLQYR	HGDACAHNATCA	HYTQLVW					
NP_872425-hCLEC18A	(76)	DSLAQLAQAARAALCGT-	PTPSLASGLWRTLVQ	GWNNQLLPAGLV	SFVEVVS LWF	FAEGQRYSHA	AEGCARNATCTHYTQLVW					
Consensus	(82)	DSLAQLA	ARAALC T TP	LAS VGNW	MLLP G SF	VEVV LWF	FAEG Y HA	AECA NATC	HYTQLVW			
	(163)	163	170	180	190	200	210	220	230	243	Section 3	
NP_853527-mCLEC18A	(162)	ATSSQLGCGRQPCFV	DOEAMEAFVCAYS	PPGNWDINGKTV	AFYKKG	TWCSLCTARV	SGCFKAWDHAGGL	CEVFRN	PCRMSC			
NP_872425-hCLEC18A	(156)	ATSSQLGCGRHLC	SAGAAIEAFVCAYS	PRGNWEVNGKTI	VYKKGAWCS	LCTASVSGCF	KAWDHAGGL	CEVFRN	PCRMSC			
Consensus	(163)	ATSSQLGCGR	C Q	AIEAFVCAYS	P GNWDINGKTI	PYKKG	WCSLCTA	VSGCFKAWDHAGGL	CEVFRN	PCRMSC		
	(244)	244	250	260	270	280	290	300	310	324	Section 4	
NP_853527-mCLEC18A	(243)	RNLGHLNISTCR	CHCPCPGYTG	RYCQVRC	SVQCVHGF	FRKEEC	SCICDVGYGGAQ	CA	TKV	FFPHTCDLRIDGDC	CFMVSFA	
NP_872425-hCLEC18A	(237)	QNHGRLNISTCR	CHCPCPGYTG	RYCQVRC	SLQCVHGR	FRREEEC	SCVCDIGYGGAQ	CA	TKV	FFPHTCDLRIDGDC	CFMVSFA	
Consensus	(244)	N G	LNISTC	CHC	PGYTG	RYCQVRC	SLQCVHG	FR	EECS	ICD	IGYGGAQ	CA
	(325)	325	330	340	350	360	370	380	390	405	Section 5	
NP_853527-mCLEC18A	(324)	DTYYGAKMKCGR	GGVLAQIESQK	VODILAFYLGR	LETTNEV	TDSD	FETKNFWIGL	TYKA	KDSFRW	TGEHQ	SFAFG	
NP_872425-hCLEC18A	(318)	DTYYRARMKCGR	GGVLAQIKSQK	VQDILAFYLGR	LETTNEV	IDSDF	ETRNFWIGL	TYK	TAKDSFRW	TGEHQ	SFAFG	
Consensus	(325)	DTYY	ARMKCO	KGGVLAQI	SO	VODILAFYLGR	LETTNEV	DSDF	ETKNFWIGL	TYK	AKDSFRW	TGEHQ
	(406)	406	420	430	440	453					Section 6	
NP_853527-mCLEC18A	(405)	QPDNQGFGNC	VEMQASAAFN	WNDQRCKTR	NRNYICQ	FAQHYSRW	EPGP					
NP_872425-hCLEC18A	(399)	QPDNQGFGNC	VEMQASAAFN	WNDQRCKTR	NRNYICQ	FAQHYSRW	EPGP					
Consensus	(406)	QPDN	GFGNC	VELQASAAFN	WNDQRCKTR	NRNYICQ	FAQ	H	SRW	PG		

3. Data in supplemental Fig1a only appears to have one data point. Can authors please clarify if this is the mean of technical triplicate, how many experiments or perform additional experiments to satisfy n number requirements for the conclusions drawn.

Answer:

We repeat this experiment with two different doses of poly (I:C) to show dose-dependent effect. We also compared the effect of poly (I:C) with other TLR ligand in cytokine expression in mRNA and protein levels (Supplementary Fig. 2) in this revised version.

4. The TLR3/dsRNA inhibitor suppresses IFN β at the same concentration in WT and CLEC18 and CLEC18A-1 cells (100uM) (Fig. 4e). This is also the case for IFN α . This result weakens the claim that C18A is in-fact a co-receptor for TLR3. Could the TLR3/dsRNA inhibitor be added to FRET experiments to definitively prove C18A/C18A-1 as co-receptor of TLR3? If not perhaps remove. The bio-layer interferometry data is much more convincing.

Answer:

- We performed FRET experiments in Figure 4f. The high concentration of TLR3/dsRNA inhibitor also disrupts the association between poly(I:C)-CLEC18A as well as poly(I:C)-CLEC18A-1. The FRET assay is in accord with what we observed in cytokine production (Figure 4e).*
- Lower concentration of inhibitor (10 μ M and 30 μ M) can only suppress cytokine release from WT, but not CLEC18A/18A-1 KI mice. However, when the inhibitor is up to 100 μ M, it also disrupts the interactions between TLR3 and poly (I:C). This observation is in accord with our claim that the CLEC18 act as a TLR3 co-receptor and can increase binding affinity between poly (I:C)/TLR3-CLEC18A complex.*

5. The confocal data in supplemental showing EEA1, TLR3 and CLEC18 co-localization is really nice and could perhaps be moved to the main figures.

Answer:

Yes, we move these supplementary figures to figure 4c & d in this revised version.

6. Figure 5b – it is unfortunate that the authors present the input blots for the immunoprecipitation studies on separate gels/western blots. It is generally expected that the Input/expression of plasmids in both the Isotype control lane and the Flag lanes be shown to confirm adequate expression of plasmid in these samples and in order to confirm a legitimate interaction. Blots in Figure 5d with deletion clones shows this appropriately.

Answer:

- 1) *Because the input gel represents the total lysates (right), and the proteins amounts in the immunoprecipitation (left) is far more than total lysates, therefore we need to separate the samples into separate gels and transfer blots, otherwise all signals in total input (right) will be far less than the immunoprecipitation (left). As far as I know, most of the papers present the total lysates and immunoprecipitation results in two sperate gels.*
- 2) *We did show the results of isotype and flag lane side by side in Figure 5b.*
- 3) *Thank you for your comment on Figure 5d that our presentation is OK. But I would like to say that the total input and IP are also developed separately in western blot analysis.*

7. Stats are needed in figure 7 for survival. Number of mice not indicated in text or figure legend. This should definitively be included prior to publication.

Answer:

We add the symbols to represent statistical significance in the figure7b now. Number of mice was shown in figure7b.

Minor

1. The labelling of human Macrophages as M-M in figure legends is very confusing for the reader. Prior to reading the legend the reader may think this data represents values from Mouse Macrophages (M-M). Can authors please explain or change this to H-M or other?

Answer:

We present the cytokines production profile of human and mice macrophages in separate panels (Figure 2, mouse BMDM) and Supplementary Figure 3 (human macrophages) to prevent confusion. ◦

2. Figure 1, Expression in mouse DC is not as convincing as in BMDM however, I appreciate the antibodies may not be great and the majority of the data in the paper is in BMDM.

Answer:

We try a couple of times and find that human CLEC18 expression in mouse DC are much weaker than that BMDM. It may due to the differential promoter activity in mouse DC and BMDM.

Reviewer #2 (Remarks to the Author):

Author tried to explained the discrepancy of TLR3 innate immune response between mice and human, and CLEC18A may causes changes of TLR3 response in human and mice. However, logics of manuscript and these results are not fully support the author`s point. In addition, control of experiments in each experiments and references are unsuitable.

The concept is certainly intriguing and is of interest. However, there are several limitations in the data as following.

1. In the introduction, explanation of mutant CLEC18A-1 is unclear. Author should what are causes by the mutation in CLEC18A function. Mutant for CLEC18A-1 is should show like “CLEC18A S339R”.

Answer:

- 1) *When we initiated the research of CLEC18, we found two CLEC18A polypeptides (NP_872425 and AAI41809) which are identical except polymorphic amino acids in CTLD and SCP domain. To understand the functions of polymorphic amino acid residue located at 339, we mutate S339 into R339 in to generate CLEC18A (S339R) mutant.*
- 2) *We agree with reviewer`s comment and change CLEC18A-1 into CLEC18A(S339R).*
- 3) *We also add a paragraph to describe difference between CLEC18A and CLEC18A(S339R mutant in the Introduction session (line 46-51).*

2. Ref 6 is not suitable. Ref 6 did not include the results of infection by H3N2

IAV.?? **Answer:** *we remove reference 6 accordingly.*

3. Explanation of Ref 8 is incorrect. Ref 8 is results of human cells. ???

Answer: *we remove reference 8 accordingly.*

4. Author mentioned that Clec18A is low expression in BMDM, however expression level of endogenous clec18a is unclear in Fig. 1.

Answer:

Mouse BMDM does not express endogenous CLEC18A mRNA (first lane from left, Fig 1 a).

5. In Fig2a and b, author included results of human macrophages. These are completely different cell line from mouse cells, therefore it is not suitable to put in the same panel and cannot discuss the production level of Ifns in between mouse and human by these results.

Answer:

We present the cytokine production profile of human (Supplementary Fig. 3) and mice (Fig. 2a) macrophages in separate figures.

6. Poly I:C treatment in vivo showed increase of inflammation in CLEC18A and mutant CLEC18A mice compared with wild type mice, however IAV treatment in these mice showed reduces of inflammation. Why the difference induced by these stimulations?

Answer:

- 1) *Poly (I:C) is able to induce inflammation, so enhanced TLR3 signaling will cause further inflammatory reactions.*
- 2) *However, TLR3 is essential for anti-viral infection (Julia Zinngrebe et al. J Exp Med. 2016), so enhanced TLR3-mediated signaling will help to clear virus infection. In addition, the antiviral genes (Ch25h, Mx1) are upregulated in CLEC18A/CELC18A(S339R) KI cells. This is why IAV infection cause less severe inflammation in the CLEC18A and CELC18A(S339R) KI mice.*

7. It is unclear why author use CLEC18A mutant mice to show difference of TLR3 responses between human and mice.

Answer:

We noticed the presence of CLEC18A(S339R) in the human genome database when we initiated the projects, so we are curious to understand whether the S339→R339 has any effect in host response to TLR3 ligand and IAV infection. Because mice do not express endogenous CLEC18A in immune cells (Figure 1), so we take this advantage to compare the differential response of CLEC18A and CLEC18A(S339R) in knock-in mice.

8. Author tried to explain the discrepancy of TLR3 innate immune response between mice and human using transgenic mice which expressed human CLCE18A. However, ectopic expression in mice to human gene does not help to explain the difference of human and mice. It is hardly compared with wild type and human gene transgenic mice. Author should compare mice gene and human gene transgenic mice. If author say the expression of Clec18a in mice is low, author should show the evidence of expression level in human and mice and use knockdown or knockout technique in human cells line.

Answer:

- 1) *As we show in Figure 1 a~c & Supplementary Fig. 1, mouse leukocytes do not express endogenous CLEC18A, neither can we detect CLEC18A in mouse lung tissue. Thus, the cytokine secretion in wild type mouse immune cells and lung tissue is only via TLR3, without the involvement of mouse endogenous CLEC18A.*
- 2) *To confirm the effect of CLEC18 in human macrophages, we further knock down endogenous CLEC18 in human macrophages (Supplementary Fig. 3). We found that*

CLEC18 knockdown in human cells only affects poly (I:C)-, but not LPS-, induced cytokine production (Supplementary Fig. 3). This observation further supports the argument that CLEC18 is the co-receptor of TLR3.

Reviewer #3 (Remarks to the Author):

Overall, their findings that human CLEC18A and CLEC18A-1 act as a co-receptor with TLR3 is particularly interesting in view of differential roles of them between human and mouse, in TLR3-mediated antiviral responses. However, data presented in this manuscript is not sufficient to support their conclusion. Furthermore, there seems to be over-interpretation throughout their manuscript. To strengthen their conclusion, many concerns listed below should be addressed.

Specific comments

1. Figure S1A, S2B and S2C should be explained in their manuscript.

Answer: *We follow the suggestion, and add explanation for what we observed in this revised manuscript. The S1A, S2B and S2C in previous version are moved to Fig. 2A, S2A, S2B, respectively, in this revised version. Description of these findings is shown in section 2 (line 256266).*

2. In figure 2A, data showing IP-10 production in response to poly(I:C) (1 microG/mL) is missing.

Answer: *We present the IP-10 production in Figure 2A*

3. In figure 2A and S1A, there seems to be some discrepancy, especially data on pro-inflammatory cytokines, they should kindly explain about that. Furthermore, they should place ELISA data but not qPCR data onto the main figure.

Answer:

1) *We follow the suggestion and present the new ELISA results on the main figure (Figure 2a in this revised version).*

2) *We describe the results in in section 2 (line 256-266) of this revised version.*

4. Figure 2a did not correspond to supplementary fig. 1a. While mRNA level of TNF-alpha showed no significant difference between WT and CLEC18 expressing cells, the protein level was increased in CLEC18 expressing cells.

Answer:

We present the new ELISA results in Figure 2a and supplementary Figure 2. No significant change in TNF- α production between WT and CLEC18 mice at protein (determined by ELISA) and RNA (determined by RT-PCR) levels.

5. They should describe about how they prepared human macrophages in [materials and methods] section.

Answer:

The description of human macrophage preparation is shown in line 96-100 of this revised version.

6. In figure 4D, to support their conclusion, they should perform statistical analysis.

Answer:

We add the symbols to represent statistical significance in the figure 4D.

7. To strengthen their conclusion, they should perform knockdown experiment using human cells.

Answer:

We perform this experiment and show the data in Supplementary figure 3. We found that knockdown of CLEC18 in human macrophages only affects poly (I:C)-, but not LPS-, induced cytokine production. This observation further supports the argument that CLEC18 is the co-receptor of TLR3.

8. In figure 6 and 7, they should show data using control (untreated) mice.

Answer:

We show the results of control mice in Figure 6 & 7.

9. In figure 1A, because the expression level of mRNA does not always reflect that of protein, they should show CLEC18 protein expression in each tissues and cells (especially immune cells). Same for figure 1B.

Answer:

We analyze the expression of endogenous CLEC18 in mice tissue and immune cells by WB (Figure 1 b&c) and flow cytometry (Supplementary Figure 1). Mouse endogenous CLEC18A is only expressed in brain, but is undetectable in heart and kidney based on western blot analysis (Fig. 1b), though low levels of mRNA are detectable in mouse kidney and heart (Fig. 1a).

We also try to detect endogenous CLEC18 by flow cytometry, but it is not detectable in T cell, B cells, neutrophils, or other myeloid cells (Supplementary Figure 1). This result is in accord with what we observed in Figure 1a&b.

3. Because their previous paper (ref. 1) also suggests that CLEC18 is secreted by cells, they should clearly refer it in their manuscript.

Answer:

We address this issue and discuss the potential role of soluble CLEC18 in the Discussion Session (line 422-431) in this revised version.

4. In figure 1C, they should clearly describe about the antibody they used in figure legend. Furthermore, in case that they used antibody against endogenous human CLEC18 protein but not against tags, they should clearly describe about the information of the antibody used.

Answer:

Description of the antibodies used in Figure 1C is shown in line 128-143 of this revised version. Because the commercially available antibody (Proteintech Catalog number: 21013-1-AP) is

unable to detect flag-tagged mouse CLEC18 (as we show in Fig. 1B) and endogenous CLEC18, we screen human phage library to identify phage clone which can recognize recombinant mouse CLEC18A. We are negotiating with Biolegend and Minipore to license this mAb now.

12. There are many typos and grammatical errors throughout the manuscript that should be corrected (e.g. [purchased] at line 101, [C57BL/J] at line 118 and [analysis] at line 133 should be [were purchased], [C57BL/6] and [analysis of], respectively.).

Answer:

We correct those typos and grammatical errors accordingly.

13. At line 133 and 134, they should not use the abbreviation [AEC] and [ATII] without prior explanation.

Answer:

We change those labeling accordingly.

14. Although they claim that CLEC18A-1 is more potent to induce interferon response, this reviewer could not find any statistical analysis between CLEC18A and CLEC18A-1.

Answer:

- 1) *We add the symbols to represent statistical significance in the figures in this revised version.*
- 2) *We rename CLEC18A-1 as CLEC18A(S339R) in this revised version.*

15. In figure 2A and 2B, they claim that CLEC18A and CLEC18A-1 increased type I interferon expression when even 1.1-1.2 fold. On the other hand, they claim that, in figure S1B, S1C and S1D, CLEC18 did not affect to ifna or ifnb mRNA expression even 2-4 fold. This is quite strange. They should perform statistical analysis in figure S1 and precisely interpret their data.

Answer:

We repeat the experiment and present the data with statistical analysis in Figure 2a and Supplementary Figure 2. Compared to WT BMDMs, higher expression of type I interferons was in CLEC18A and CLEC18A(S339R) cells after poly (I:C) stimulation. In contrast, other TLR ligand has no obvious effect to upregulate cytokine production.

16. In figure 4C, statistical analysis seems to be strange. The difference in ifnb between WT (pIC 0) and WT (pIC 30) was significant, but that between WT (pIC 0) and WT (pIC 10) was not significant. If the figure is correct, latter also seems to be significant. They should confirm the method of statistical analysis throughout their manuscript.

Answer:

I think the reviewer may misunderstand what we present in Figure 4c (changed to Figure 4e in this revised version). In this experiment, all groups were incubated with the same concentration of the poly (I:C), but with the different concentration of TLR3/dsRNA complex inhibitor as indicated. We re-analyze data by one-way ANOVA method.

17. In figure 5D, the means of [+] and [-] is not clear.

Answer:

We describe the means of [+] in legend (line 584-585) in this revised version.

18. In figure 5D, it is interesting to investigate which domain of CLEC18 is involved in its binding to poly (I:C).

Answer:

In Figure 5D, we want to identify CLEC18 domain responsible for association with TLR3. Based on our BLI binding study, CLEC18 interacts with poly (I:C) via CTLD domain (Table II).

Reviewers' comments:

Reviewer #1 (Remarks to the Author):

I commend the authors in their efforts to address all of the reviewers comments and perform additional experiments.

I am satisfied that authors have answered the concerns of all three reviewers.

I would suggest minor amendments:

In supplemental figure 3 for the knockdown of CLEC18A by two different shRNA can authors please include data demonstrating sufficient knockdown.

In the relevant text and discussion regarding comments on the induction of phospho-TBK1 and phospho-IRF3 between CLEC18A and CLEC18A(S339R) expressing BMDM and WTs(Figure 4g). It is clear that the induction of p-TBK1 and p-IRF3 is enhanced in CLEC18A(S339R) compared to WT cells. However this does not appear as clear for the results presented for CLEC18A compared to WT. This simply adds to the overall argument in the discussion that the ROSA-CLEC18A(S339R) allele may be more beneficial following viral infection than CLEC18A and therefore warrants further investigation. I think the overall text should reflect this. Or do the others have other blots for CLEC18A and pTBK1/p-IRF3? n numbers are not indicated in the figure legend for these experiments.

The manuscript reads well but requires additional proof reading for grammar and typos prior to publication. For example lines [36-37], [234-239], [256-266], [96 - 100].

Reviewer #2 (Remarks to the Author):

In this manuscript, Huang et al, mentioned mainly two points. The one is human CLEC18A support TLR3 function to enhance inflammatory response, and the other is hCLEC18A mutation at S339R further enhance inflammatory response by regulating TLR3-poly IC binding. Author greatly improved in the revised manuscript, however there are several discrepancies.

To investigate hCLEC18A function in TLR3 dependent pathway, author showed ectopic human CLEC18A expression in mouse cells causes TLR3 response, human genes expression in mouse cells is unlikely to be occurred in natural condition. Although author newly included knockdown experiment by using human cells, author should show the role of CLEC18A and CLEC1A(S339R) function in human cells. For example, Author should express CLEC1A(S339R) in knock down cells and investigate inflammatory and ifn genes expression and poly IC binding.

Author showed the role of mutation in S339R, however several Type I IFN and inflammatory genes expression are comparable between 18A and 18A(S339R) in Figure 2 and 3, and hCLEC 18A(S339R) mice only showed protection from IAV infection in Figure 7, and hCLEC 18A(S339R) mice showed enhanced inflammation in Figure 6. There are several discrepancy in the role of S339R in those figures.

Reviewer #3 (Remarks to the Author):

The authors adequately addressed all the concerns raised by this reviewer. I think the revised manuscript is suitable for the publication.

Reviewers' comments:

Reviewer #1 (Remarks to the Author):

1. In supplemental figure 3 for the knockdown of CLEC18A by two different shRNA can authors please include data demonstrating sufficient knockdown.

Answer:

The knock-down efficiency of two siRNAs is shown in the supplementary figure 3(a) of this revised manuscript, and the legend is highlighted in yellow.

2. In the relevant text and discussion regarding comments on the induction of phospho-TBK1 and phosphor-IRF3 between CLEC18A and CLEC18A(S339R) expressing BMDM and WTs(Figure 4g). It is clear that the induction of p-TBK1 and p-IRF3 is enhanced in CLEC18A(S339R) compared to WT cells. However this does not appear as clear for the results presented for CLEC18A compared to WT. This simply adds to the overall argument in the discussion that the ROSA-CLEC18A(S339R) allele may be more beneficial following viral infection than CLEC18A and therefore warrants further investigation. I think the overall text should reflect this. Or do the others have other blots for CLEC18A and pTBK1/p-IRF3? n numbers are not indicated in the figure legend for these experiments.

Answer:

1) *We agree the comment and address the beneficial role of CLEC18A(S339R) vs CLEC18A in Discussion session (line 415-429) of this revised manuscript.*

2) *We add the 'n' numbers in line 603 of this revised manuscript.*

3. The manuscript reads well but requires additional proof reading for grammar and typos prior to publication. For example lines [36-37], [96 - 100], [234-239], [256-266],

Answer:

We revised these sentences and corrected the typos in this revised manuscript. All the newly added sentences are highlighted in yellow. lines [36-37], [93-100], [244-250], [265-274].

Reviewer #2 (Remarks to the Author):

In this manuscript, Huang et al, mentioned mainly two points. The one is human CLEC18A support TLR3 function to enhance inflammatory response, and the other is hCLEC18A mutation at S339R further enhance inflammatory response by regulating TLR3-poly IC binding. Author greatly improved in the revised manuscript, however there are several discrepancies.

1. To investigate hCLEC18A function in TLR3 dependent pathway, author showed ectopic human CLEC18A expression in mouse cells causes TLR3 response, human genes expression in mouse cells is unlikely to be occurred in natural condition. Although author newly included knockdown experiment by using human cells, author should show the role of CLEC18A and CLEC18A(S339R) function in human cells. For example, Author should express CLEC18A(S339R) in knock down cells and investigate inflammatory and ifn genes expression and poly IC binding.

Answer:

We perform this experiment accordingly, and the results are shown in supplementary figure 4. Because primary macrophages become fragile after siRNA transfection, cells died after following plasmid transfection to express human CLEC18A and CLEC18A(S339R). Thus, we transect siRNA to human HT1080 cells to knockdown endogenous CLEC18, followed by transfecting plasmids to overexpress CLEC18A and CLEC18A(S339R). The description of this experiment is shown in the materials and methods (line 122-133), and text (line 278-284) (highlighted in yellow).

2. Author showed the role of mutation in S339R, however several Type I IFN and inflammatory genes expression are comparable between 18A and 18A(S339R) in Figure 2 and 3, and hCLEC18A(S339R) mice only showed protection from IAV infection in Figure 7, and hCLEC18A(S339R) mice showed enhanced inflammation in Figure 6. There are several discrepancy in the role of S339R in those figures.

Answer:

We summarize the differential responses between CLEC18A and hCLEC18A(S339R) allele and discuss the potential role of CLEC18A(S339R) allele in host defense against various microbial infections in the DISCUSSION session of this revised manuscript (line 415-429).

a) The results of Figure 2 and 3 are from in vitro studies.

We did find differential response of hCLEC18A and hCLEC18A(S339R) BMDMs (Fig. 2) and epithelial cells (Fig. 3) to viral infections in vitro, and compared with cytokine production and mice survival rates in vivo (Fig. 7). We found CLEC18A(S339R) is more potent than CLEC18A to respond to viral infections and poly. (I:C) stimulation.

b) Figure 6 is based on the stimulation of poly (I:C) in vivo.

Because CLEC18A and hCLEC18A(S339R) enhance host response to poly (I:C), so ROSA-CLEC18A and ROSA-CLEC18A(S339R) mice display more severe inflammatory reactions than WT mice in tissue section (Fig. 6a). These observations are consistent with our observations that CLEC18A and CLEC18A(S339R) enhance cytokine productions to poly (I:C) in vitro.

c) Figure 7 is based on IAV infection in vivo.

These results are in accord with our claim that hCLEC18A and hCLEC18A(S339R) enhance the expression of interferon and ISG to suppress virus infection. Therefore, less cell infiltration and lower virus titer are found in ROSA-CLEC18A and ROSA-CLEC18A(S339R) mice.

Reviewer #3 (Remarks to the Author):

The authors adequately addressed all the concerns raised by this reviewer. I think the revised manuscript is suitable for the publication.

Answer: Thank you very your nice comment

REVIEWERS' COMMENTS:

Reviewer #1 (Remarks to the Author):

The authors have addresses all of this reviewers concerns. The manuscript has also been corrected for typos and the discussion improved as requested.

I believe the manuscript is now suitable for publication

Reviewer #2 (Remarks to the Author):

Authors address the concern. I think the revised manuscript is suitable for the publication.